# Regionalizing the Sea-level Budget With Machine Learning Techniques

Carolina M.L. Camargo[1,2], Riccardo E.M. Riva[2], Tim H.J. Hermans[1,2], Eike M. Schütt[3], Marta Marcos[4], Ismael Hernandez-Carrasco[4], and Aimée B.A. Slangen[1]

[1]NIOZ Royal Netherlands Institute for Sea Research, Department of Estuarine and Delta Systems, Yerseke, The Netherlands
[2]Delft University of Technology, Department of Geoscience and Remote Sensing, Delft, The Netherlands
[3]Department of Geography, Kiel University, Kiel, Germany
[4]Mediterranean Institute for Advanced Studies (IMEDEA), Spanish National Research Council-University of Balearic Islands (CSIC-UIB), Esporles, Spain

**Correspondence:** Carolina Camargo, carolina.camargo@nioz.nl

**Abstract.** Attribution of sea-level change to its different drivers is typically done using a sea-level budget approach. While the global mean sea-level budget is considered closed, closing the budget on a finer spatial scale is more complicated due to, for instance, limitations in our observational system and the spatial processes contributing to regional sea-level change. Consequently, the regional budget has been mainly analysed on a basin-wide scale. Here we investigate the sea-level budget at sub-basin scales, using two machine learning techniques to extract domains of coherent sea-level variability: a neural network approach (Self-organizing Maps, SOM) and a network detection approach ($\delta$-MAPS). The extracted domains provide more spatial detail within the ocean basins and indicate how sea-level variability is connected among different regions. Using these domains we can close, within 1-sigma uncertainty, the sub-basin regional sea-level budget from 1993-2016 in $100\%$ and $76\%$ of the SOM and $\delta$-MAPS regions, respectively. Steric variations dominate the temporal sea-level variability and determine a significant part of the total regional change. Sea-level change due to mass exchange between ocean and land has a relatively homogeneous contribution to all regions. In highly dynamic regions (e.g., Gulf Stream region) the dynamic mass redistribution is significant. Regions where the budget cannot be closed highlight processes that are affecting sea level but are not well captured by the observations, such as the influence of western boundary currents. The use of the budget approach in combination with machine learning techniques leads to new insights into regional sea-level variability and its drivers.

## 1 Introduction: The sea-level budget

Sea-level change is one of the major challenges of the coming centuries for coastal communities worldwide (Fox-Kemper et al., 2021). Global mean sea-level change has been rising at a rate of 1.6 mm yr$^{-1}$ since 1900, and 3.3 mm yr$^{-1}$ since 1993 (Frederikse et al., 2020). However, sea level does not change uniformly: it displays strong spatial and temporal variations (Hamlington et al., 2020). Ocean dynamics, land ice mass changes and associated gravitational effects, vertical land movement and the inverse barometer effect are some of the processes responsible for these regional differences (e.g., Stammer et al., 2013;

Slangen et al., 2017). Understanding the regional variability of the processes driving sea-level change is critical for improving our understanding of its causes, constraining sea-level projections, and to better prepare for the impacts of climate change.

The attribution of sea-level change to its different drivers is typically done using a sea-level budget approach (Chambers et al., 2017; Cazenave et al., 2018). For 1993-2018, about one third of the observed rate of global mean change can be attributed to thermal expansion of the oceans, while the rest is due to the effect of water and ice mass exchanges between land and ocean (Frederikse et al., 2020). Since the observed rate of sea-level change matches, within uncertainties, with the sum of the contributions of the various sources, the global mean sea-level budget for the period 1993-2018 is considered to be closed (Cazenave et al., 2018; Frederikse et al., 2020; Chen et al., 2020; Barnoud et al., 2021). However, locally attributing the drivers of sea-level change for this same period still leads to large differences between the total measured change and the sum of the contributions (e.g., Slangen et al., 2014; Royston et al., 2020). This is partly due to the spatial resolution of the current observational systems of the sea-level budget components and of the processes in question, which still limits the closure of the budget on a local spatial scale, for instance on a 1 degree resolution (Royston et al., 2020). Consequently, the regional sea-level budget has mainly been analysed on a basin-wide scale (e.g., Purkey et al., 2014; Frederikse et al., 2018, 2020; Royston et al., 2020) and has not been closed on sub-basin scales consistently for the entire world. The sea-level budget has also been analysed for individual coastline stretches characterized by coherent variability (Rietbroek et al., 2016; Frederikse et al., 2016, 2017; Dangendorf et al., 2021), and at individual tide gauges (Wang et al., 2021).

The basin-scale sea-level features extracted by Thompson and Merrifield (2014) have been frequently used in regional sea-level budget studies (Purkey et al., 2014; Frederikse et al., 2018, 2020; Royston et al., 2020). Although these publications have made significant advances in understanding the regional sea-level change, the basin scale is still too large to really understand the causes of local variations. In this manuscript, we argue that understanding the spatial structure of contemporary sea-level change is a key point to move towards a budget with finer spatial resolution. By identifying smaller physically coherent regions, some of the effects of small scale variability can be removed, allowing to close the budget at a sub-basin scale. Machine learning techniques, such as complex and neural networks, can be used to identify such spatial structures, determining the ideal resolution and regions of common sea-level variability and change. While machine learning methods have widely been used in oceanography (e.g., Richardson et al., 2003; Liu et al., 2006; Hernández-Carrasco and Orfila, 2018; Sonnewald et al., 2019; Falasca et al., 2019, 2020; Novi et al., 2021), only few examples analysing sea-surface height can be found (e.g., Liu et al., 2016; MA et al., 2016; Sonnewald et al., 2018). Here, we apply two unsupervised machine learning techniques –self-organizing maps (SOM) and $\delta$-MAPS– to extract coherent spatial features (domains) in sea-level change observations.

In this study we use the extracted domains to analyse the sea-level budget on a sub-basin scale during the satellite altimetry period (1993-2016), by using state-of-the-art estimates of sea-level change and its components. We limit our analysis to 2016 because of the temporal span of the hydrological models used to obtain the landwater storage contribution to sea-level change. Additionally, instrumental problems (e.g., in Argo salinity data and satellite drifts) have raised questions about the performance and closure of the global mean sea-level budget after 2016 (Chen et al., 2020; Barnoud et al., 2021; Cazenave and Moreira, 2022). We hypothesize that by investigating the budget in covariant and physically coherent regions, we can resolve the discrepancies (i.e., close the budget) that appear in an increased-resolution sea-level budget (e.g., 1x1 degree).

## 2 Data and methods

In this section we introduce the data sets used for each of the different components of the sea-level budget (Section 2.1). We also describe the trend and budget analysis (Section 2.2) and introduce the machine learning techniques used to extract coherent regions (domains) of sea-level variability and change (Section 2.3).

### 2.1 The components of the regional sea-level budget

For the budget, we compare the total observed sea-level change $\eta_{total}$ to the sum of the drivers of sea-level change $\eta_{drivers}$:

$$\eta_{total} = \sum \eta_{drivers}, \tag{1}$$

where $\eta$ stands for the rate of sea-level change.

Total sea-level change ($\eta_{total}$) can be measured by tide gauges and satellite altimeters. Satellite altimeters measure geocentric or absolute change ($\eta_{geo(sat)}$), that is, the sea surface height in relation to the reference ellipsoid (Gregory et al., 2019). On the other hand, tide gauges measure sea surface height in reference to a terrestrial landmark ($\eta_{rel(TG)}$), registering the relative sea-level change. The latter is affected by vertical land motion (VLM) due to, for instance, land subsidence and tectonics (Wöppelmann and Marcos, 2015), while geocentric sea level can not differentiate if the change is either from the solid Earth or the ocean. The relationship between geocentric and relative sea-level change is:

$$\eta_{total} = \eta_{geo(sat)} = \eta_{rel(TG)} + VLM. \tag{2}$$

From hereon, when we use $\eta_{total}$, we are referring to the geocentric sea-level change derived from satellite altimetry (Figure 1a). We use multi-mission gridded Level-4 data from 4 distribution centers: CMEMS (CMEMS, 2022), JPL MEaSUREs (Zlotnicki et al., 2019), SLcci (SLcci, 2022) and CSIRO (CSIRO, 2022). All of these products use the same reference ellipsoid model (GRS80/WGS), and have a monthly temporal resolution, except for JPL MEaSUREs time series which provides sea surface height data every 5 days and was averaged into monthly means. All data is regridded to a $1°\text{x}1°$ map, selected within $66°\text{S}$ to $66°\text{N}$ of latitude, and combined into an ensemble mean, to avoid systematic errors. We apply a glacial isostatic adjustment (GIA) correction to the altimetry data from ICE-6G VM5a (Argus et al., 2014; Peltier et al., 2015), by removing the rate of change of the geoid (i.e., Drad + Dsea) from the trends.

Sea-level change expresses changes in the volume of the ocean. These can be caused by changes in the ocean density, mass or area. Density-driven changes, known as steric sea-level changes, are caused by variations in the ocean temperature and salinity (Gill and Niller, 1973; MacIntosh et al., 2017). All sea-level variations not driven by density changes are known as manometric sea-level change (Gregory et al., 2019). Thus, Equation 1 can be rewritten to:

$$\eta_{total} = \sum \eta_{drivers} = \eta_{SSL} + \eta_{MAN}, \tag{3}$$

where $\eta_{SSL}$ and $\eta_{MAN}$ refer to steric and manometric sea-level change, respectively.

For steric sea-level change ($\eta_{SSL}$, Figure 1c), we use the estimates of Camargo et al. (2020), which are based on fifteen different ocean temperature and salinity data sets down to 2000m depth, using Argo floats (Roemmich and Gilson, 2009;

Gaillard et al., 2016; Li et al., 2017; Lu et al., 2019), multiple in-situ observations (Ishii and Kimoto, 2009; Guinehut et al., 2012; Cabanes et al., 2013; Good et al., 2013; Gaillard et al., 2016; Ishii et al., 2017; Cheng et al., 2019; Szekely et al., 2019) and ocean reanalyses (Blockley et al., 2014; Maclachlan et al., 2015; Storto and Masina, 2016; Garric and Parent, 2017; Carton et al., 2018; Zuo et al., 2019). We complement this data with the deep ocean steric estimate of Purkey et al. (2019, updated from Purkey and Johnson (2010)).

Manometric sea-level change ($\eta_{MAN}$), also referred to as the bottom pressure term (Gregory et al., 2019), can be further divided into (i) $\eta_{GRD}$, sea-level change due to the Gravitational, Rotational and viscoelastic Deformation (GRD) response of the Earth to water and ice mass exchanges between land and ocean, and (ii) $\eta_{DSL}$, the dynamic redistribution of ocean mass due to ocean circulation, atmosphere and ocean bottom pressure changes as a result of the steric change of the oceans (Landerer et al., 2007), following:

$$\eta_{MAN} = \eta_{GRD} + \eta_{DSL}. \tag{4}$$

The GRD component ($\eta_{GRD}$, Figure 1d) reflects how the mass loss of continental ice stored in glaciers and ice sheets and variations in land water storage affect sea level. The GRD effect can be split between responses due to contemporary changes, and due to the response of the Earth to the last ice age, known as post-glacial rebound or GIA. The integrated response of the GRD effect over the oceans, i.e. the global mean, is known as barystatic sea-level change ($\eta_{BSL}$, Gregory et al., 2019). For the GRD component, we use the estimates from Camargo et al. (2022), which includes the geocentric sea level response to changes on the Antarctic and Greenland ice sheets, glaciers and terrestrial water storage. These are based on a suite of different estimates of land mass change, and computed solving the sea-level equation following Farrell and Clark (1976) and Slangen et al. (2014).

The dynamic component ($\eta_{DSL}$, Figure 1e) refers to mass changes driven by bottom pressure changes, that is, the redistribution of mass that was already in the oceans. Note that, by our definition, the dynamic sea-level change ($\eta_{DSL}$) is part of the ocean dynamic change ($\Delta_\zeta$, Gregory et al., 2019), the latter also including the effect of local steric anomalies ($\eta'_{SSL}$). That is, the dynamic term here is the residual of the sterodynamic sea-level change with the steric contribution removed (Gregory et al., 2019). $\eta_{DSL}$ is computed from the sea-surface height of five ocean reanalyses (Table 1), by first removing the time-varying global mean from the sea-surface height, and then by removing the local steric anomaly. This procedure is done in each ocean reanalysis individually, and we then combine the five estimates into an ensemble. We acknowledge that this method introduces some circularity to the budget analysis: the reanalysis, used to obtain $\eta_{DSL}$, assimilates satellite sea-surface height, and in the budget analysis we compare this estimate with satellite sea-surface height ($\eta_{total}$). Compared with the $\eta_{DSL}$ estimated from Gravity Recovery and Climate Experiment Satellite (GRACE, Tapley et al., 2004), $\eta_{DSL}$ sea-level trends from 2005-2015 agree on large scale patterns and magnitude of dynamic changes (Figure A1). Note that our budget components do not incorporate GRACE mass changes over the oceans, hence it is an independent estimate for validation. More detail on the estimation and validation of $\eta_{DSL}$ is given in Appendix A.

Finally, Equation 3 can be rewritten as:

$$\eta_{total} = \eta_{SSL} + \eta_{GRD} + \eta_{DSL}, \tag{5}$$

such that the total observed sea-level change (Figure 1a) can be compared with the sum of the components (Figure 1b). The ensemble mean of each term of Equation 5, used throughout this manuscript for the sea-level budget analysis, is shown in Figure 1, where $\eta_{total}$ is the geocentric sea-level change from satellite altimetry, corrected for the inverted barometer and GIA ($\eta_{GIA}$) effects; $\eta_{SSL}$ is the full-depth steric sea-level change; $\eta_{GRD}$ is the contemporary ocean mass redistribution due to the land-ocean mass exchange, already corrected for $\eta_{GIA}$ effects; and $\eta_{DSL}$ is the mass redistribution due to purely ocean dynamics. A summary of the budget components and data sets sources is given in Table 1. Note that all the used data sets have been homogenised to a monthly temporal resolution and a $1°x1°$ spatial resolution.

**Table 1.** Summary of the sea-level budget components and data sources used in this manuscript.

| Symbol | Name | Description | Reference |
| --- | --- | --- | --- |
| $\eta_{total}$ | Observed change | Total sea-level change from satellite altimetry | Ensemble of CMEMS (CMEMS, 2022), JPL MEa-SUREs (Zlotnicki et al., 2019), SLcci (SLcci, 2022) and CSIRO (CSIRO, 2022) |
| $\eta_{SSL}$ | Steric expansion | Full depth density-driven sea-level change due to ocean temperature and salinity variations | Camargo et al. (2020) and Purkey and Johnson (2010) |
| $\eta_{GRD}$ | Mass change | Contemporary ocean mass redistribution due to the land-ocean mass exchange | Camargo et al. (2022) |
| $\eta_{DSL}$ | Dynamic change | Mass redistribution due to purely ocean dynamics | Ensemble of SODA (Carton et al., 2018), C-GLORS (Storto and Masina, 2016), GLORYS (Garric and Parent, 2017), FOAM-GloSea (Blockley et al., 2014; Maclachlan et al., 2015) and ORAS (Zuo et al., 2019) |

## 2.2 Computing Trends and Uncertainties

Our sea-level budget includes the comparison of sea-level time series, trends and associated uncertainties. We assume that sea-level trends are the sum of a deterministic model (including annual and semi-annual signals) and stochastic noise (temporal uncertainty). We use the software Hector (Bos et al., 2013) to compute the trends and the associated 1-sigma uncertainty for each of the budget components. Following Bos et al. (2014); Royston et al. (2018); Camargo et al. (2020, 2022), we test 8 different noise-models to describe the auto-correlation between the residuals of the regression. Using the Akaike and Bayesian information criteria (Akaike, 1974; Schwarz, 1978), we select the best performing noise-model at each grid cell. More information on the noise-model analysis can be found in Camargo et al. (2020, 2022). For the GRD component, in addition to the temporal uncertainties, we also consider the spatial, structural and intrinsic uncertainties (Camargo et al., 2022).

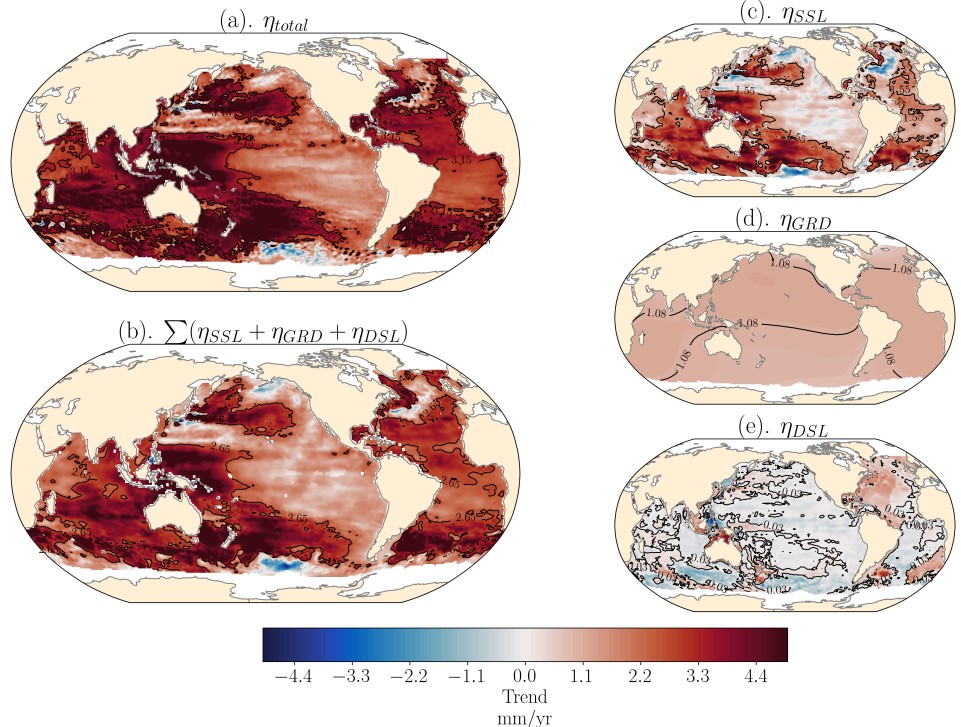

**Figure 1.** Regional sea-level trends for 1993-2016 (mm/yr) for (a) altimetry; (b) sum of sea-level components: (c) full-depth steric, (d) GRD effect and (e) dynamic sea-level change. Black contour line indicates global mean sea-level change.

Note that, unlike for the identification of the domains (Section 2.3), the time series used to estimate trends and uncertainties include seasonality and global mean trends.

We assume independence of the terms, and sum the trends linearly and uncertainties in quadrature. For each sea-level domain we take the area-weighted spatial average of the time series, trend and uncertainties. Performance of the sea-level budget is evaluated by (i) the magnitude of the residual, (ii) the Pearson's correlation coefficient ($r$) between the altimetry time series and the budget components, and (iii) the normalized root mean squared error (nRMSE). nRMSE measures the distance between the true value, in this case altimetry, and the modeled value, in this case the sum of the budget components. Contrary to $r$, nRMSE closer to 0 indicates better performance.

## 2.3 Clustering Techniques

To answer our research questions, we must first identify regions with similar sea-level variability. To do so, we use two different machine learning pattern detection algorithms, one based on an neural network approach, Self-organizing Maps (SOM), and one based on a deep network detection method, $\delta$-MAPS. The methodological differences in these two techniques leads to different patterns of sea-level change in terms of geographical location, region size, and ocean coverage. Hence, by using both

methods, we can (i) find prevailing sea-level modes, (ii) compare the patterns and sea-level budget for the different methods and (iii) balance out the advantages and disadvantages of using a single method. Both methods are used to reduce the dimensionality of the data, transforming high-dimensional input data into low-dimensional features (Liu et al., 2006; Falasca et al., 2020). For both clustering techniques we use $1°x1°$ monthly satellite altimetry time-series (CMEMS, 2022) from 1993-2019 as input. Note we use a longer time-series than the ones for the budget analysis, as longer time-series can resolve better the temporal variability. However, additional tests (not shown) showed that the clustering is not strongly affected by the extra 3 years of data. We pre-process the input data by removing the global mean trend, seasonality and by applying a spatial Gaussian filter of 300km half-width to remove small scale variability. Note that, after the domains identification, for the budget analysis, global mean trend, seasonality and small scale variability are included in the time series. Smaller seas, such as the Mediterranean, Baltic, Black and Caspian seas have been removed from the data prior to the clustering.

### 2.3.1 Self-organizing Maps (SOM)

SOM (Kohonen, 1982) is a feature extraction and classification method based on an unsupervised neural network (Liu et al., 2006), which was demonstrated to be more powerful than conventional feature extraction methods (e.g., Liu and Weisberg, 2005). The ability of SOM to extract patterns of sea level variability from satellite altimetry data has been shown in previous works (e.g., Hardman-Mountford et al., 2003; Liu et al., 2008; Iskandar, 2009; Liu et al., 2016; Weisberg and Liu, 2017; Nickerson et al., 2022). To analyse sea level data, SOM can be applied either in the spatial domain, focusing on the characteristic spatial patterns, or in the time domain, focusing on the characteristic time series (Liu et al., 2016). The latter results in regionalizing the sea-level variability, and is pursued here to analyse global sea level data. We use the MatLab SOM toolbox (Vesanto et al., 2000), and follow Liu et al. (2006) and Hernández-Carrasco and Orfila (2018) to choose the parameters. We apply the SOM algorithm in the time domain in order to extract the spatial patterns, herein referred to as domains, based on coherent temporal sea-level variability. Before initizaling the SOM, the 3D input data (time,lat,lon) is concatenated to 2D (time, latxlon; Richardson et al., 2003; Liu et al., 2016), and normalized to have unit variance. The network is initialized linearly, based on the first two principal components of the time series, and trained in a batch mode, that is, at each step of the training process, all input data vectors are simultaneously used to update the network. Training is performed over 10 iterations, which is necessary to stabilize and converge the network, while avoiding overfitting of the SOM (Liu et al., 2006). We use the 'Epanechikov function' as a neighborhood function, which returns the most accurate SOM patterns, a hexagonal lattice, and a neighborhood radius (determining the radius of cells that are updated during the training process) of 2 cells at the beginning, decreasing linearly to 1 during the training process. We tested different SOM parameters, and verified that this combination gave the smallest quantification errors by computing the averaged Eulerian distance between each data input vector and the best matching unit (BMU). SOM domains do not need to be geographically contiguous, that is, different non-connected regions can be assigned to a single domain. Initially, the strong sea-level variability of the Equatorial Pacific Ocean dominated the clustering, hindering pattern identification in the Atlantic Ocean (Supplementary Figure B7). To overcome this issue we perform the clustering analysis on the Atlantic and Indo-Pacific Ocean basins separately. We select a map size of 3x3 neurons (i.e., neural network nodes) in each basin, leading to a total of 18 domains. Using different map sizes (e.g., Supplementary Figure B7) led to more

"patchy" results, hence we used map size of 3x3 neurons as a compromise between the amount of detail and the interpretability of the domains.

### 2.3.2 $\delta$-MAPS

$\delta$-MAPS (Fountalis et al., 2018) is a complex network methodology which reduces the spatiotemporal dimensionality of a field by identifying regions (domains) with similar dynamics and their connectivity (Bracco et al., 2018; Falasca et al., 2020). Here we focus only on the domains identification (dimensionality reduction) function of the $\delta$-MAPS method. $\delta$-MAPS domains are spatially continuous (i.e., grid cells need to be physically connected to the be clustered in the same domain) and are potentially overlapping regions that have a highly correlated temporal activity (Falasca et al., 2019). Formally, each input grid cell is associated with a time series, including the K nearest neighbors, based on the haversine distance (angular distance between two points on a sphere). The local homogeneity, defined as the average Pearson cross-correlation between a grid cell and its K-neighbors, is computed and tested against a threshold value $\delta$. If the local homogeneity is greater than $\delta$, with a statistical significance level of 0.1, then the grid cell is considered a core, which then is expanded to identity domains (Fountalis et al., 2018; Falasca et al., 2019; Novi et al., 2021). Each domain expands to adjacent cells, as long as the local homogeneity continues to be higher than $\delta$. To choose the optimal neighborhood size K, we follow a heuristic approach, testing K values from 4 to 25 following Falasca et al. (2019). As in $\delta$-MAPS not every grid point needs to belong to a domain (in contrast to SOM), we then choose the K-value taking into account the amount of unclustered cells (i.e., the one with most of the ocean belonging to domains). We also use the normalized mutual information (NMI) matrix (Falasca et al., 2019) to identify the K-value with high NMI for it and its neighboring K-values, meaning that the results are less sensitive to the chosen K-value. These parameters led to the use of K = 5.

## 3 Identifying Domains of Sea-Level Variability

Both clustering methods successfully reduce the dimensionality of the input data, despite the higher number of domains identified by $\delta$-MAPS (Figure 2). SOM identified 18 coherent domains, with a domain area varying from 3.84 to 34.51 million km$^2$, and an average and total size of 17.61 and 316.90 million km$^2$. $\delta$-MAPS identified 92 coherent domains, with a domain area varying from 0.03 to 24.15 million km$^2$, with average and total size of 2.53 and 242.01 million km$^2$. Despite the methodological differences, we find that prominent sea-level features are clustered in a similar way by SOM and $\delta$-MAPS (Figure 2). Some of the patterns identified can be linked with known oceanic patterns, as we will discuss below. However, we note that covariability does not imply a common forcing, and that some patterns may be statistically separated or grouped without a clear physical reason. It is also important to note that these clustering methods do not account for auto-correlation in time, that is the time lag in the progression of a signal across the ocean basin. Since we use monthly data, signals that propagate faster than a month (typically barotropic) will be more clearly correlated in our clustering. On the other hand, slower propagating signals, such as the first baroclinic mode, will lose correlation in space and will not be represented in the identified domains.

The central Pacific domain, where the variability is dominated by El Niño Southern Oscillation (ENSO) events, covers a similar region in both methods. The 'ENSO-tongue', starting from the coast of Peru and Ecuador and spreading west until the central Pacific, is identified by both methods (SOM domain 12 (pink), $\delta$-MAPS domain 45 (light green)). The Western Tropical Pacific Ocean (WTPO), influenced by ENSO and the Pacific decadal oscillation (PDO), is also identified as a single domain by both methods (SOM domain 16 (light green), $\delta$-MAPS domain 89 (light brown)). The WTPO domains matches with the region of significant spatial correlation between steric and coastal sea-level found by (Dangendorf et al., 2021) for West Australia. In the SOM clustering, the WTPO domain incorporates the Leeuwin Current (Western Australia, Pattiaratchi and Siji, 2020) in the Indian Ocean, which is affected by waves travelling through the Tropical Australasian Seas (Feng et al., 2004). While this connection is not captured by $\delta$-MAPS, the coherence along the western coast of Australia is featured in a single domain ($\delta$-MAPS domain 92, light pink). The Kuroshio Extension region is also identified in both methods (SOM domain 10 (brown), $\delta$-MAPS domain 88 (brown)), reflecting how strong boundary currents influence the sea-level variability. Another example is the North Atlantic, which has similar clustering in both methods, especially in the domain south of Greenland (SOM domain 9 (light purple), $\delta$-MAPS domain 33 (purple)), which is marked by decadal-scale sea-level change reflecting the strength and shape of the wind-driven Subpolar Gyre and the Atlantic Meridional Overturning Circulation (Chafik et al., 2019). Within these domains, density anomalies are known to flow southward from the Labrador Sea into the Subpolar Gyre through coastally trapped waves (Dangendorf et al., 2021). Another region identified in both methods is the Northwestern European Shelf (SOM domain 8 (purple), $\delta$-MAPS domain 66 (grey)), which is part of a domain that extends along the whole western European coast, continuing down to the Canary islands and well into the Atlantic. This connection could be related to the hypothesis that coastally trapped waves and longshore winds cause a coherent region of sea-level variability from around the latitude of the Canary Islands up to the Norwegian Sea (Calafat et al., 2012; Chafik et al., 2019; Hughes et al., 2019; Hermans et al., 2020; Dangendorf et al., 2021). These features are in a separate $\delta$-MAPS domain (53, green) than the Northwestern European Shelf. It is important to note that coherent features smaller than 300km are not captured in the domains because of the spatial filtering applied before the clustering analysis.

As SOM domains do not need to be contiguous, possible pseudo-teleconnections between different ocean regions (within the Atlantic and Indo-Pacific Ocean basins) come out of the analysis. For example, areas adjacent to the 'ENSO-tongue' domain, both north and south are clustered together in domain 18 (light blue) or in domain 15 (moss green), indicating how the ENSO signal is propagated through the Pacific, possibly through coastally trapped waves (Hughes et al., 2019) in the coastal domains (15), or via atmospheric teleconnections. However, not every region classified into the same SOM domain results from a clear connection. For example, SOM domain 17 (blue) groups the ocean adjacent to South Africa, the region below the Kuroshio Extension (offshore of Taiwan) and a region south of Australia and New Zealand. Another example is SOM domain 7 (salmon-pink), which implies a connection between the Atlantic Caribbean Sea and the west part of the South Atlantic Gyre (capturing parts of the Brazil Current). These regions have been classified together because they have a similar behaviour in terms of sea-level variability, but probably different forcing. Further investigation, with ocean currents, ocean-atmospheric oscillations and ocean waves are necessary to explore and quantify the physical connection behind these patterns.

Unlike SOM, every $\delta$-MAPS domain is assigned a unique number and not every pixel needs to be clustered (Figure 2a, white regions). Consequently, this method yields a larger number of domains with smaller size, while avoiding pseudo-teleconnections. The dominant sea-level modes are clear on $\delta$-MAPS clustering, reflecting the influence, for example, of ENSO and western boundary currents on sea level. For example, the entire Caribbean (domain 87 (brown)) and Gulf of Mexico (domain 82 (red)) is in a single domain, highlighting the similarity in that region. The same goes for the Equatorial Atlantic (domain 86 (light purple)), the ENSO region (domains 45, 89 and 62 (light green, light brown and light brown, respectively)), and the Kuroshio current (domain 88 (brown)).

As shown in Royston et al. (2020), the components of the sea-level budget have a similar spectral power to the total observed sea-surface height of altimetry between wavelengths of approximately 3,000 and 10,000 km. The clustering techniques applied here not only reduce the dimensionality of the data, but also average out sea-level variability in regions of coherent variability, being ideal for a regional budget analysis (next section).

## 4 The Regional Sea-Level Budget on Different Spatial Scales

### 4.1 Sea-Level Trend Budget Closure

We investigate the trends of the sea-level budget on different spatial scales, from a finer (1x1 degree) to coarser scale ($\delta$-MAPS and SOM domains; Figure 3). The residuals (i.e., the difference between the total sea-level change and the sum of the components) decrease towards a coarser spatial scale: for 1 degree, they range from -8.2 to 21.1 mm yr$^{-1}$, while for $\delta$-MAPS they range from -1.2 to 3.8 mm yr$^{-1}$, and for SOM from 0.1 to 0.7 mm yr$^{-1}$. This shows an improvement of the budget closure (i.e., total and sum of components agree within uncertainties) by using the pattern detection algorithms: the budget closes in all 18 SOM domains (100% of SOM ocean area), in 70 out of 92 of the $\delta$-MAPS domains (94% of $\delta$-MAPS ocean area (229.9 million km$^2$)) and in 72% of the grid cells in the 1 degree budget (75% of the ocean area) (Figure 3). There is a clear relation between spatial scale of the region considered for the trend, and the residuals of the budget (see also Figure B5). The good closure in the 1 degree budget is likely an artefact of the large uncertainties of the observations, which on a local scale can be up to 18.9 mm yr$^{-1}$ (see Supplementary Figure B2). This is in line with Royston et al. (2020), who found that local biases of steric estimates together with the resolution limitation of GRACE observations over the oceans hinder the budget closure at 1 degree resolution. When the regional domains based on SOM and $\delta$-MAPS are considered, the uncertainties show a fivefold reduction compared to the 1 degree resolution, reaching up to 3.6 mm yr$^{-1}$ and an average value of 1.6 mm yr$^{-1}$ (Supplementary Figure B2), while the budget still closes.

Consequently, there is a better match between the total observed rate of sea-level change with the sum of the components for the clustered regions (scatter points in Figure 3, right column), with a reduction in the spread of the scatter points and moving closer to the 1:1 line (black dashed line) for the coarser resolutions. The dashed pink lines in Figure 3 indicate the half-width of the 95% confidence interval of the uncertainty of the residuals, showing a slightly larger width for 1 degree, and a smaller one for SOM and $\delta$-MAPS. Even when the components uncertainties (grey error bars) are considered, the scattered values are mostly within the width of the 95% confidence interval for the SOM domains, confirming the improvement of the budget for

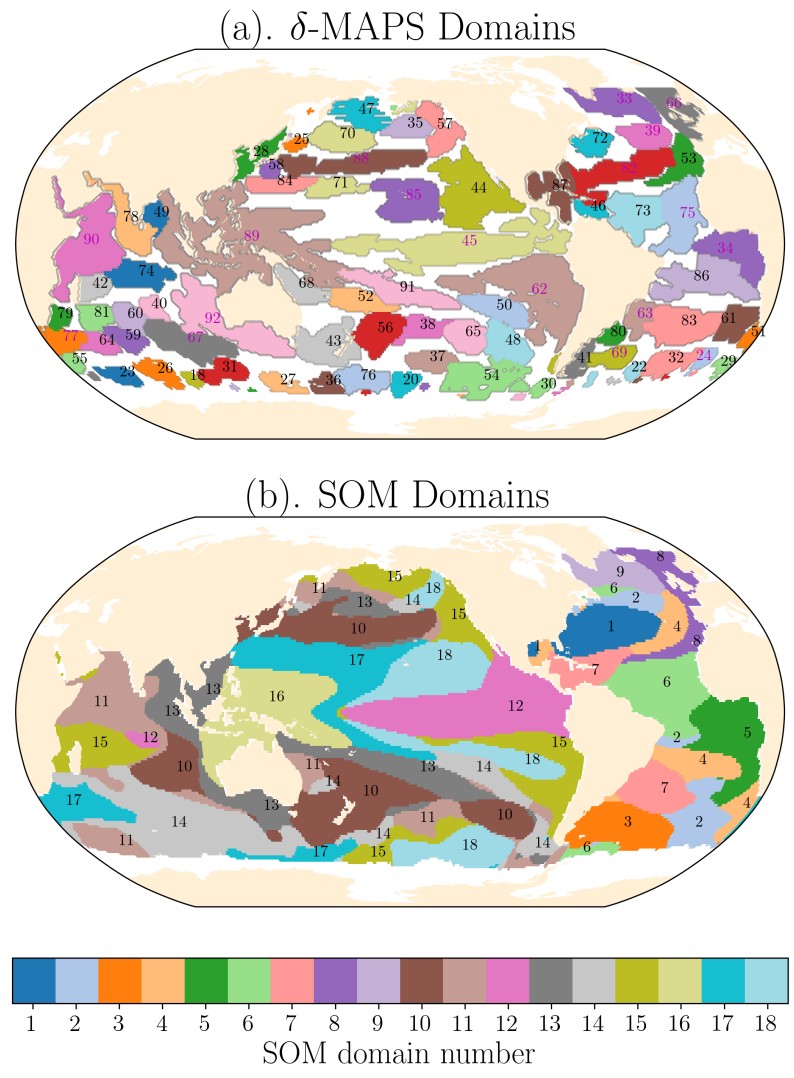

**Figure 2.** Domains of coherent sea-level variability. (a): δ-MAPS method (92 domains); (b): Self-organizing Maps (SOM) method (18 domains). Numbers indicate the domain code, domain names are given through the main text and in Supplementary Table B1. δ-MAPS domains with codes in magenta indicate selected domains for Figure 4. For visibility, small domains have not been labeled. Given the large number of domains, δ-MAPS has a repeating color pallet, but since δ-MAPS domains need to be continuous, repeated colors do not indicate the same domain. White regions in δ-MAPS indicate incoherent regions, which were not incorporated in any domain.

this case. There is a strong linear correlation between the total and the sum of the drivers, with Pearson's r varying from 0.81 for the 1 degree budget and $\delta$-MAPS to 0.98 for SOM. The RMSE also decreases for the coarser scales, from 1.01 mm/yr for the 1 degree budget to 0.47 mm yr$^{-1}$ for the SOM domains.

The altimetry trends are generally larger than the sum of the sea-level change drivers, as indicated by the positive residuals and scatter points above the 1:1 line on Figure 3. This is true for more than half of the $\delta$-MAPS domains and for all SOM domains except one: SOM domain 9 (South of Greenland) is marked by a negative residual, that is, the sum of the drivers is larger than the observed altimetry trend. Several $\delta$-MAPS domains, such as Southwest of Australia (domains 92 and 67), Southeast Pacific (domains 37 and 54), Gulf Current (domain 82) and Brazil-Malvinas confluence zone (domains 80 and 69),
also have a negative residual. This might indicate a larger temporal variability or regime shifts in this region, or might be due to the ocean dynamics contribution, such as the effect of the Subpolar Gyre around the south of Greenland (Chafik et al., 2019), as we will see in the next section (Section 4.2).

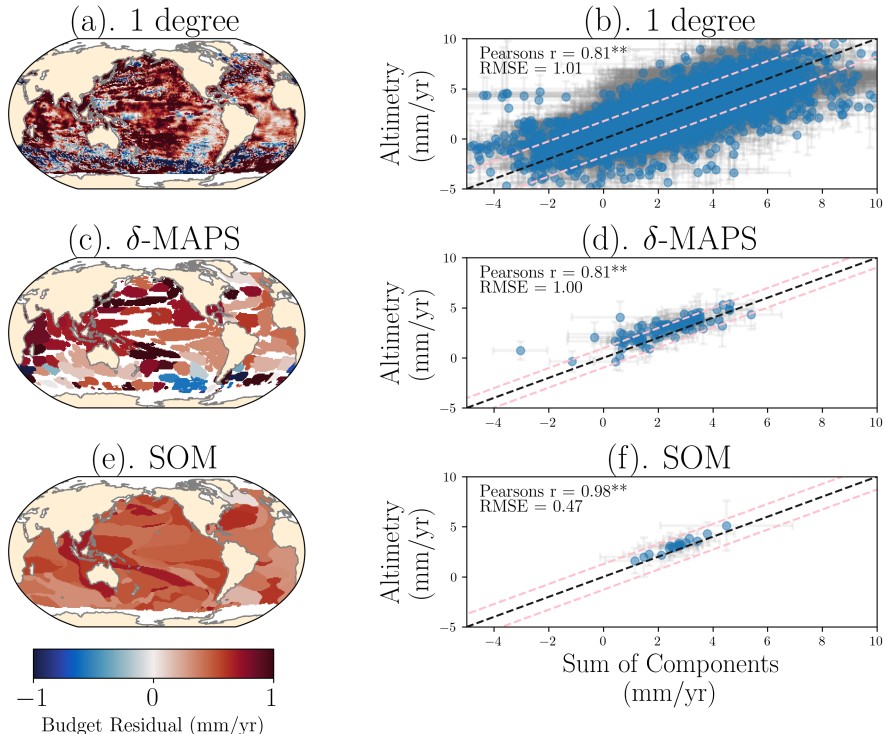

**Figure 3.** Sea-level budget residuals (maps(a,c,e)) and comparison between total sea-level change (y-axis) and sum of components (x-axis) (scatter plots (b,d,f)) for 1 degree (a,b), $\delta$-MAPS domains (c,d) and SOM domains (e,f). Gray lines indicate the uncertainties (1-sigma) of the components. In the scatter plots every point indicates one region (grid in case of 1degree), pink dashed lines indicate the half-width of the 95% confidence interval of the residuals uncertainty, grey error bars indicate the component uncertainty. ** indicates that coefficient is statistically significant (p-value <0.01).

## 4.2 Explaining the Sea-Level Budget Contributions

In this section, we investigate which components dominate the trend and temporal variability in each of the different domains. For comparison and discussion purposes, we choose 18 $\delta$-MAPS domains (magenta numbers, Figure 2b) located close to the 18 SOM domains. Trends for all $\delta$-MAPS domains are available online as an interactive map (see caption Figure 4).

As shown previously, we find a good match of total observed sea-level change and the sum of components (Figure 4a,b, green stars and purple triangles, respectively) for all SOM and $\delta$-MAPS domains. The largest budget uncertainties, considering both altimetry and the sum of components, is seen in the WTPO domain (SOM 16, $\delta$-MAPS 89). These uncertainties may be related to: (i) poor performance of standard altimetry products in these shallow regions; (ii) poor Argo float coverage in the region (Kleinherenbrink et al., 2017), influencing both the steric and dynamic components; and (iii) large internal variability due to ENSO events in this region, which may contribute to large temporal uncertainties in the steric and altimetry components (Kleinherenbrink et al., 2017; Wagner and Böning, 2021). This region is also within the Indian-south Pacific basin (Thompson and Merrifield, 2014), which was the only basin in which the regional budget from 2005-2015 could not be closed (Royston et al., 2020).

The GRD component (Figure 4, blue) has a relatively comparable contribution to all regions, contributing about $1.5$ mm yr$^{-1}$ of sea-level rise. The dynamic and steric components, however, show a strong regionally varying contribution (Figure 4, red and yellow, respectively). For example, for SOM ($\delta$-MAPS) domains 10 (88), 13 (92), 14 (67) and 16 (89), more than $50\%$ of the total trend is due to steric variations. On the other hand, for SOM domains 1 and 18 and $\delta$-MAPS domains 39, 45 and 62, the steric trend explains less than $20\%$. The dynamic component shows a small contribution for most of SOM domains, and in some domains even a negative trend (e.g., SOM domain 11, 12 and 14). An exception is the Gulf Stream domain (SOM 1, $\delta$-MAPS 82), where almost half of the total trend is explained by the dynamic component. This dominance of the dynamic component reflects the influence of the strong western boundary current on sea level in this region. The south of Greenland domain (SOM 9, $\delta$-MAPS 33) also includes a relatively large dynamic contribution, with a trend of $0.49 \pm 0.21$ mm yr$^{-1}$, reflecting the influence of the Subpolar Gyre in this region. The dynamic component also has a significant contribution to other $\delta$-MAPS domains, such as domains 24, 69, 39, 66 and 67. Domain 67, located southwest of Australia, shows a large negative dynamic trend, which can be related to the influence of the West Australian Current.

Regarding the temporal evolution (Figure 4c,d and Supplementary Figure B4), both SOM (solid lines) and $\delta$-MAPS (dashed lines) time series show a similar behaviour. The steric component dominates the temporal sea-level variability, with a good match to the altimetry. The time series of 'ENSO-tongue' domain (SOM 12 and $\delta$-MAPS 45, Figure 4d) shows the clear response of sea level to ENSO, with peaks coinciding with strong ENSO events, such as the El Niño of 1997 and 2015. The prominent contribution of the dynamic component to the total trend in the Gulf Stream domain (SOM 1 and $\delta$-MAPS 82) is not reflected in the time series (Figure 4c). Hence, while the dynamic component has a significant impact on the overall change, it does not contribute to the seasonal to interannual sea-level variability. This is true for all other domains (Supplementary Figure B4), except for SOM ($\delta$-MAPS) domain 2 (24) and 18 (85), where we find a better match between the dynamic and altimetry time series.

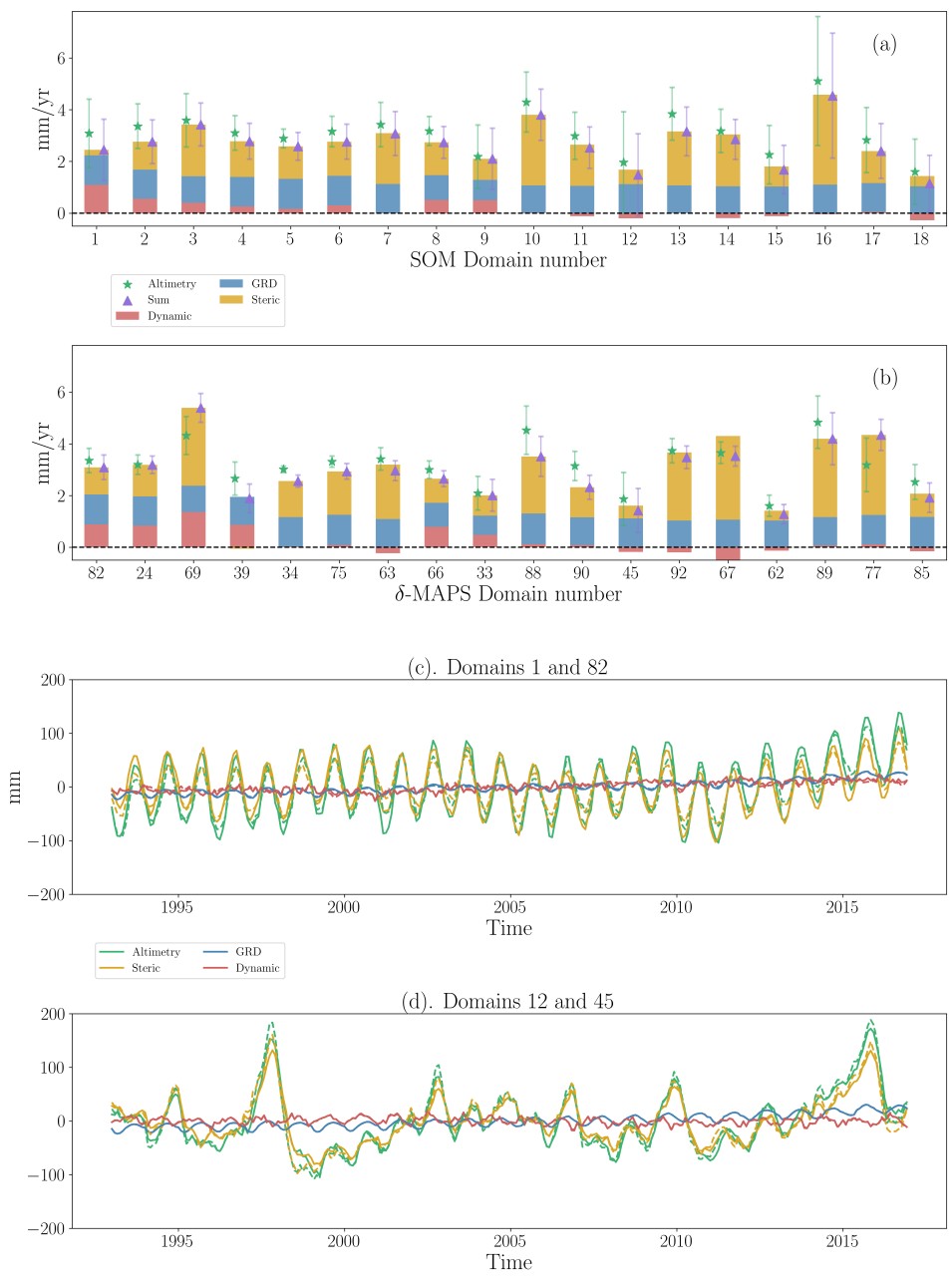

**Figure 4.** Sea-level budget trends (mm/yr) for (a) SOM and (b) $\delta$-MAPS domains, and (c,d) time series for two example domains, where solid and dashed lines indicate SOM and $\delta$-MAPS time series, respectively. Location of each domain is shown in Figure 2 (domain numbers in magenta for $\delta$-MAPS). For comparison $\delta$-MAPS domains are matched to the SOM domains, for example SOM domain 12 to $\delta$-MAPS domain 45. Bar plot for all other $\delta$-MAPS domains can be found in Supplementary Figure B3. Error bars indicate the 1-sigma uncertainty of the trend. Time series for all SOM domains and for the 18 $\delta$-MAPS domains in (b) are shown in Supplementary Figure B4. An interactive budget map is available at https://carocamargo.github.io/resources/regional-SLB-domains/ for both SOM and $\delta$-MAPS.

### 4.3 Sea-Level Budget Performance

Here, we investigate the closure of the budget considering (i) the components included in the budget, (ii) the size of the domains and the clustering method, and (iii) the data sets used for each component.

To illustrate the performance of the budget considering the domains used and the components included in the budget, we show how the Pearson's correlation coefficient ($r$) and the normalized root mean squared error (nRMSE) change when these factors vary (Figure 5). This figure firstly shows that the budget closure improves when more components are included in the budget. While we get a poorer performance when only considering the dynamic or the GRD component, the budget with only steric already performs relatively well. The improved correlation and lower RMSE with the steric component is not surprising;

the seasonal cycle is predominantly steric. The budget performance is enhanced by the addition of the dynamic and GRD components, shown by the narrowing of the box-and-whiskers plot.

The figure also shows an improvement of the budget closure for $\delta$-MAPS and SOM domains, in relation to the 1 degree resolution, regardless of the budget combination. There are two possible reasons why a coarser spatial resolution leads to decreasing uncertainties and a better budget closure: (i) the spatial scale of the process itself, as changes in long-term sea level

typically occur on a coarser resolution than 1 degree; and/or (ii) there is a mismatch in the exact location between the sum of the components and altimetry observations on a finer spatial scale, resulting from the limited resolution of the observations, compared to a coarser scale when such mismatches are partially averaged out. Additionally, the averaging of more samples leads to a smaller standard error. However, the measurement errors between altimetry and the sum of components will only compensate each other if they are uncorrelated in the spatial scale being analysed. The relationship between the spatial scale

of the domains and the performance of the budget is further confirmed in Figure B5, which shows how the residual of the budget decreases when larger regions are considered. Note however, that simply upscaling the resolution of the observations – i.e, considering 2x2 or 5x5 degrees blocks – does not have the same effect on budget performance as the domains derived by machine learning (Supplementary Figure B6): there is demonstrated added value of considering regions that are physically coherent, rather than artificial blocks, for the budget analysis. That is, spatially averaging over areas of similar variability

reduces the unexplained variance of the observations.

When it comes to the data sets of the different sea-level change drivers, sea-level budget studies often use the ensemble mean of several data sets for each component (e.g., Cazenave et al., 2018), or they compute a range of budget combinations, by varying the data set for each component, to find the combination that returns the best budget closure (e.g., Gregory et al., 2013). The latter approach can result in a budget closure for the wrong reasons (Royston et al., 2020). On the other hand, while

the ensemble mean approach may reduce the systematic biases of using individual data sets (Storto et al., 2017), it may also hinder the real variability of the process being analysed (Rougier, 2016). Alternatively, the budget can be analysed with the data sets closest to the ensemble means, according to the RMSE analysis, which retains the true variability of an individual data set (Rougier, 2016; Royston et al., 2020).

All the results presented so far were computed using the ensemble means for each component, considering 15 steric, 5

dynamic, 4 barystatic and 4 altimetry data sets. Considering all single data sets plus the ensemble of each component we can

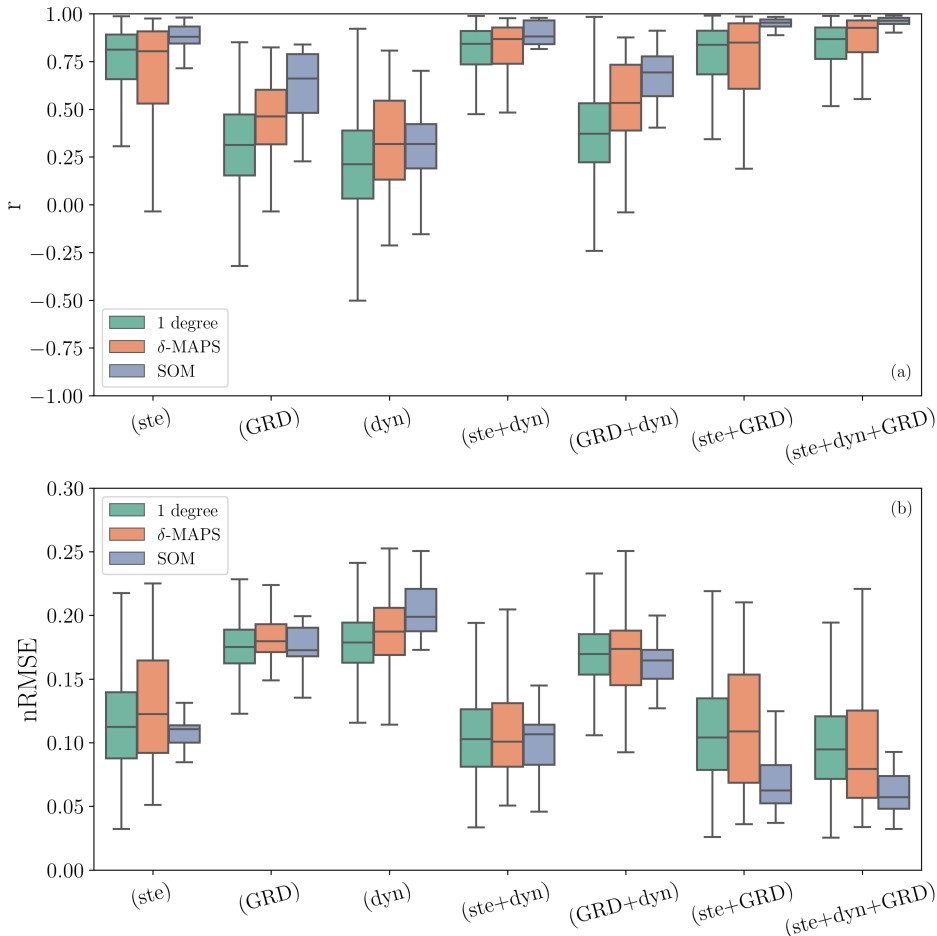

**Figure 5.** The effect on budget closure for different component combinations and spatial resolutions. (a) Pearson's correlation coefficient ($r$) and (b) normalized root mean squared error (nRMSE), in mm yr$^{-1}$, between total sea-level (altimetry) and the different components included in the budget (x-axis), for the different spatial resolutions (1 degree in green, $\delta$-MAPS in red and SOM in blue). Boxes represent the quartiles of the distribution, extending from the lower to upper quartile values of the data, with a line at the median, while the whiskers (not error bars) show the full distribution.

obtain 2400 possible budget combinations (16x6x5x5). To illustrate the dependence of the budget closure on the data set used, we now also discuss the residuals of each SOM domain considering all 2400 possible data set combinations (Figure 6). The residual value shows a large spread for the different budget combinations, ranging from about -2 to 2 mm yr$^{-1}$, and 33% of the combinations would result in non-closure of the budget (i.e., the sum of the components does not match with the altimetry values, indicated in red).

The residuals of the ensemble combinations (used throughout this study, and indicated by the blue filled squares in Figure 6) are comparable with the residuals of the combinations using the data sets with the smallest RMSE to the ensemble mean (indicated with purple filled triangles in Figure 6). With the exception of the domains 14 and 16, we see that the ensemble and the RMSE combination have a similar residual value. This indicates that the closure of the budget is not an artefact of the data set choice.

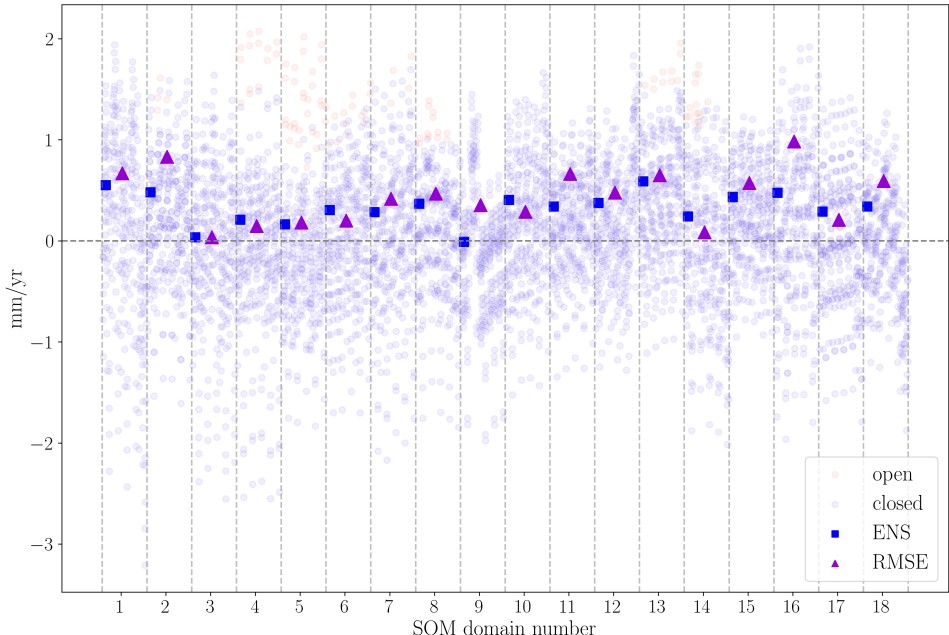

**Figure 6.** Budget residuals (mm/yr) for all possible data sets combinations for every SOM domain (separated by vertical dashed lines). The ensemble mean combination (used for the main analysis) is indicated with squares. The RMSE combination, that is, the budget combination using for each component the individual data set with smallest RMSE in relation to the ensemble mean, is indicated with triangles.

## 5   Discussion & Conclusions

Sea-level budget assessments are important tools for understanding the processes driving sea-level change, for detecting temporal changes in sea-level and its components, for identifying missing contributions to the budget, and for validating and constraining climate models used in sea-level projections (Cazenave and Moreira, 2022). In particular, understanding the pro-

cesses on a finer spatial scale is essential for local sea-level projections and coastal management planning. In this study, we investigated the regional sea-level budget for 1993-2016 on a global scale.

Regional sea-level budget closure tends to be difficult due to the complex physical processes acting on different spatial scales. To overcome this spatial resolution issue, we applied a neural network approach, SOM, and a deep-network detection method, $\delta$-MAPS, to identify domains of coherent sea-level variability (Figure 2). Note however, that the coherent patterns will be different on whether total sea surface height or the individual components (steric, dynamic, GRD) are considered. Hence, depending on the purpose of the study, it is important to first remove the unwanted components from total sea-surface height and then perform the clustering. The identified patterns reflect, among others, the influence of natural internal climate modes (Han et al., 2017), such as ENSO, PDO and NAO. This indicates the potential of using machine learning and pattern detection algorithms, such as SOM and $\delta$-MAPS, to isolate the effects of natural climate modes from anthropogenic forcing on sea-level change. The domains also suggest how sea-level variability may be transferred between ocean regions. For example, the Northwestern European Shelf SOM domain extends south down to the Strait of Gilbratar, possibly reflecting how coastally trapped waves propagate sea-level variability into the North Sea (Calafat et al., 2013; Dangendorf et al., 2014; Hughes et al., 2019; Hermans et al., 2020; Dangendorf et al., 2021). Additionally, highly energetic ocean regions, such as the Kuroshio current, the Gulf Stream and the Malvinas confluence zone, are also extracted as single features, matching the spectrum of sea-level variability in those zones (Hughes and Williams, 2010).

Compared with the basin regions of Thompson and Merrifield (2014), we have identified more and smaller domains, especially in the Southern Hemisphere. This means our domains can provide an additional level of spatial detail compared to ocean basins, while remaining large enough to provide a consistently closing regional sea-level budget. Using the domains identified with SOM and $\delta$-MAPS, we presented a regional sea-level budget assessment on an average scale of about $5 \cdot 10^6 km^2$, with the largest regions about $30 \cdot 10^6 km^2$. The performance of the budget improves from finer (1 degree resolution) to coarser scale (SOM domains), with a residual spread of 0.6 mm yr$^{-1}$ for SOM compared to 29.2 mm yr$^{-1}$ for 1 degree resolution. We also showed that the budget closes better when all components (steric, dynamic and GRD) are included, highlighting the importance of including the deep steric and dynamic contributions to regional sea-level change. Despite the large uncertainties at a regional scale (compared to the global mean) (Royston et al., 2020), we were able to identify dominant drivers in most domains. The $\delta$-MAPS regions where the budget cannot be closed highlight processes that are affecting sea level but are not well captured by the observations, such as the influence of western boundary currents and dynamic processes (e.g., the Malvinas Confluence zone). They may also be related to the quality of global data sets in continental shelves and close to the coast, or to instrumental noise.

The GRD component has a relatively homogeneous contribution, independent of the domain, in agreement with Frederikse et al. (2020). The steric contribution dominates the seasonal and interannual variability, and results in the prevailing sea-level trend in most domains, especially for domains in the southern hemisphere and equatorial regions. The dynamic component is important in some regions, particularly in the Gulf Stream domain. The domains where the dynamic component plays an important role coincide with the coastal polygons of Rietbroek et al. (2016) where a large part of the budget could not be explained solely by the sum of steric and land-ocean mass exchange. Hence, our analysis sheds light on the unexplained vari-

410 ance of previous sea-level budget studies. Note that the sea-level analysis in coastal regions is more challenging (Dangendorf et al., 2021), since some of the dominant coastal ocean dynamics are not properly represented in the global data sets (Liu and Weisberg, 2007).

Here we showed that pattern detection techniques based on machine learning, such as SOM and $\delta$-MAPS, are powerful approaches for identifying and understanding features of global sea-level change and variability. The domains identified in this

research highlight that different ocean regions are interconnected, revealing how large-scale circulation controls regional sea level. These domains are not only a good starting point for a regional sea-level budget analysis, but also have the potential to separate natural and anthropogenic forcings of sea-level change in a detection and attribution approach, building on previous work (e.g., Marcos and Amores, 2014; Slangen et al., 2014, 2016). Future work may include multiple linear regressions with climate modes to explore this potential. Additionally, these domains can also be used for coastal sea-level reconstructions (e.g.,

as Dangendorf et al. (2021)) and for pattern scaling in sea-level projections (Bilbao et al., 2015).

*Code and data availability.* Sea-level trends and scripts used for the budget analysis are available at 10.5281/zenodo.7007330 and https://github.com/carocamargo/SLB. Interactive maps of the sea-level budget are available at https://carocamargo.github.io/resources/regional-SLB-domains/

## Appendix A: Dynamic Sea-level Change Estimation and Validation

The dynamic redistribution of mass due to ocean circulation and atmospheric redistribution effects is known as dynamic sea-level change ($\eta_{DSL}$ Landerer et al., 2007; Gregory et al., 2019). $\eta_{DSL}$ refers to mass changes driven by bottom pressure changes, that is, the redistribution of mass that was already in the oceans, and includes mass exchange at any point by mass redistribution, by wind stress and by non-linear interaction due to density changes. Note that, by our definition, the dynamic sea-level change ($\eta_{DSL}$) is part of the ocean dynamic sea-level change ($\Delta_\zeta$, Gregory et al., 2019), the latter also including the

effect of local steric anomalies ($\eta'_{SSL}$). When the ocean dynamic component ($\Delta\zeta$) is considered together with the global mean steric sea-level change ($\overline{\eta_{SSL}}$), then it is known as sterodynamic sea-level change ($\eta_{SDSL}$, Gregory et al., 2019; Dangendorf et al., 2021; Wang et al., 2021). By decomposing the steric component in a global mean (denoted with the overline bar) and local anomaly component (denoted by the prime symbol), we can write the sterodynamic equation as:

$$\eta_{SDSL} = \Delta\zeta + \overline{\eta_{SSL}} = \eta_{DSL} + \eta'_{SSL} + \overline{\eta_{SSL}} \tag{A1}$$

To obtain $\Delta_\zeta$, we use the sea-surface height of 5 ocean reanalysis data sets (SODA (Carton et al., 2018), C-GLORS (Storto and Masina, 2016), GLORYS (Garric and Parent, 2017), FOAM-GloSea (Blockley et al., 2014; Maclachlan et al., 2015) and ORAS (Zuo et al., 2019)). As ocean reanalyses are mass conserving (Griffies and Greatbatch, 2012), the sea-surface height of a reanalysis does not include the GRD component, but it does include the steric effect. We acknowledge that this method introduces some circularity to the budget analysis: the reanalysis, used to obtain $\eta_{DSL}$, assimilates satellite sea-surface height,

and in the budget analysis we compare this estimate with satellite sea-surface height ($\eta_{total}$). Following Wang et al. (2021) we

compute ocean dynamic sea-level change by removing the time-varying global mean from the reanalysis' sea surface height:

$$\Delta\zeta = \eta_{rea} - \overline{\eta_{rea}}. \tag{A2}$$

Since we are interested purely in the dynamic part of $\Delta_\zeta$, that is, the dynamic sea-level change ($\eta_{DSL}$), we must remove the steric local anomaly ($\eta'_{SSL}$) as:

$$\eta_{DSL} = \Delta\zeta - \eta'_{SSL} = \Delta\zeta - (\eta_{SSL} - \overline{\eta_{SSL}}), \tag{A3}$$

where the steric estimate has been computed with the ocean temperature and salinity of the respective reanalysis. We then compute the ensemble mean of the 5 dynamic estimates.

To validate our estimate of $\eta_{DSL}$ , we compare it with $\eta_{DSL}$ estimated from Gravity Recovery and Climate Experiment Satellite (GRACE, Tapley et al., 2004). GRACE measures total mass changes, which can be used to derive estimates of mano-metric sea-level change over the oceans, that is the change in response to both the dynamic ocean mass redistribution ($\eta_{DSL}$) and to mass redistribution due to the land-ocean mass exchange ($\eta_{GRD}$) (Chambers et al., 2004; Royston et al., 2020). We use GRACE mass concentrations (mascons) products over the oceans from two different processing centres: RL06 from the Center for Spatial Research (CSR, Save et al., 2016; Save, 2020) and RL06 v02 from the Jet Propulsion Laboratory (JPL, Watkins et al., 2015; Wiese et al., 2019). In order to obtain the $\eta_{DSL}$, we then remove the GRD patterns obtained for the same data sets by Camargo et al. (2022). Note that we use GRACE dynamic sea-level change for validation purposes, but not in our budget analysis, as this data set only starts in 2002.

Qualitatively, $\eta_{DSL}$ obtained from GRACE (Figure A1a) and from ocean reanalysis (Figure A1b) agree on large scale patterns and magnitude of dynamic changes, despite local differences (Figure A1c). The main differences are in the region surrounding Indonesia and Japan, related to the signature of the 2004 Sumatra (Indonesia) and 2011 Tokohu (Japan) mega-thrust earthquakes (Chen et al., 2007; Ghobadi-Far et al., 2020) on GRACE observations. To a lesser extent, we also see the effect of the 2010 Maule (Chile) earthquake and tsunami (Ghobadi-Far et al., 2020). Another strong divergence is seen in the South Atlantic, where the positive trends of GRACE are not represented in the reanalysis, possibly suggesting that a source of dynamic sea-level change is not well parameterized in the reanalysis. Alternatively, this divergence might also be an artefact of the GRACE spherical harmonic solutions and low-degree corrections.

**Appendix B: Supplementary Figures and Tables**

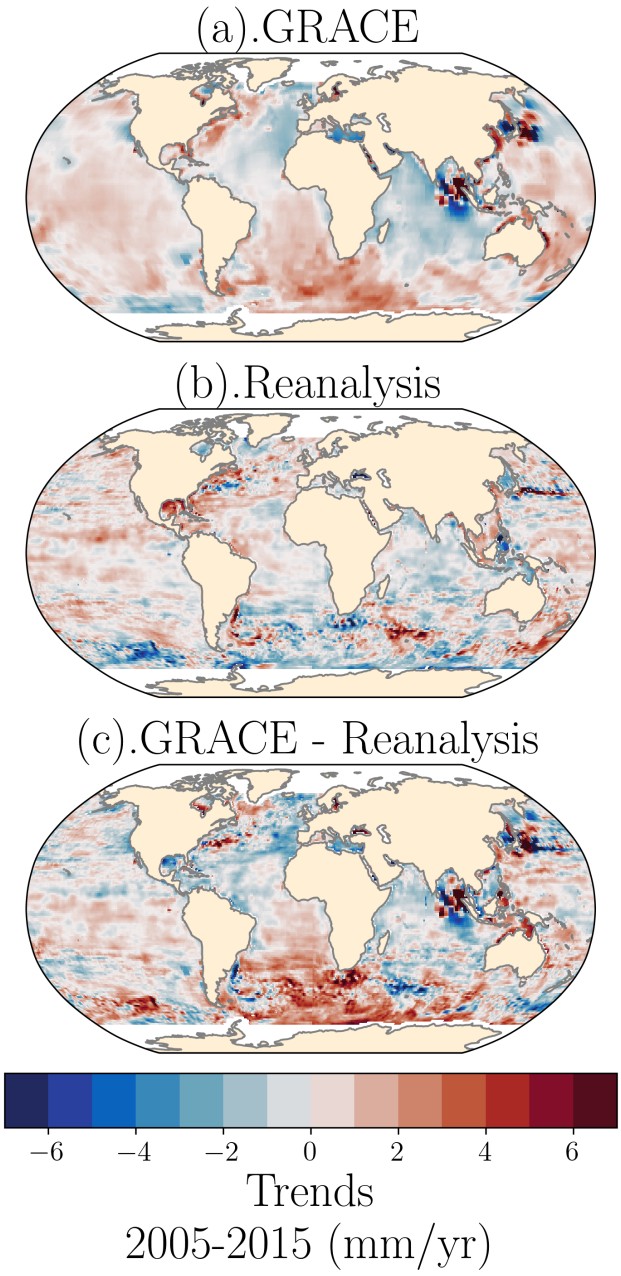

**Figure A1.** Dynamic sea-level change ($\eta_{DSL}$) estimated from (a) GRACE (average of JPL and CSR mascons), (b) ensemble of ocean reanalysis, (c) difference between GRACE and reanalysis.

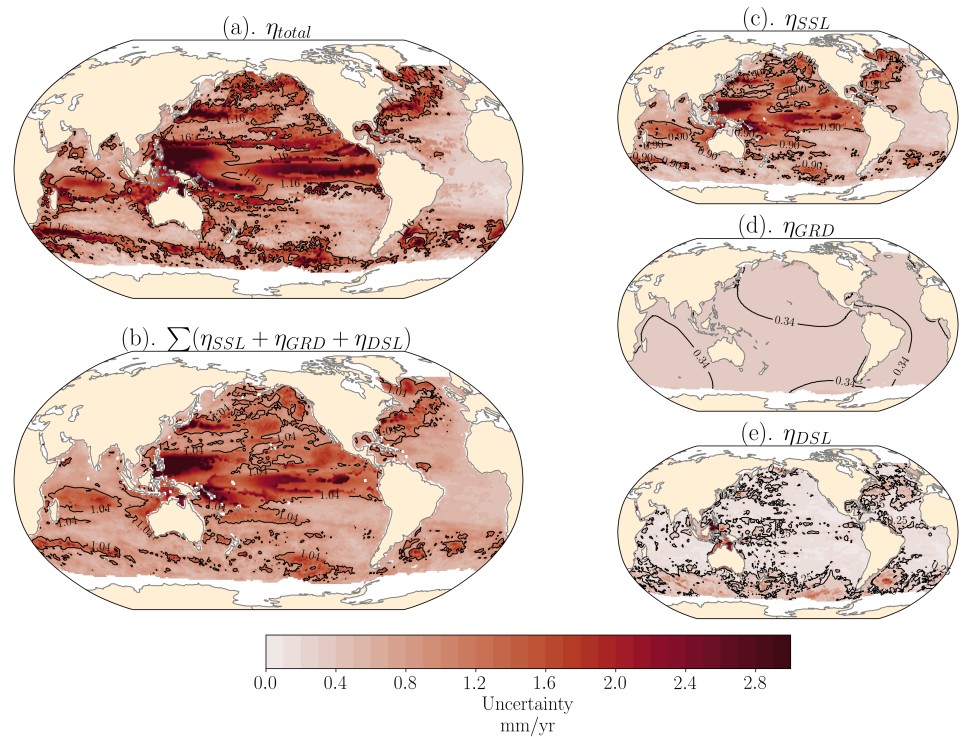

**Figure B1.** Uncertainties of the regional sea-level trends (Figure 1) for 1993-2016 (mm/yr) for (a) altimetry; (b) sum of sea-level components: (c) full-depth steric, (d) GRD effect and (f) dynamic sea-level change. Black contour line indicates global mean sea-level change.

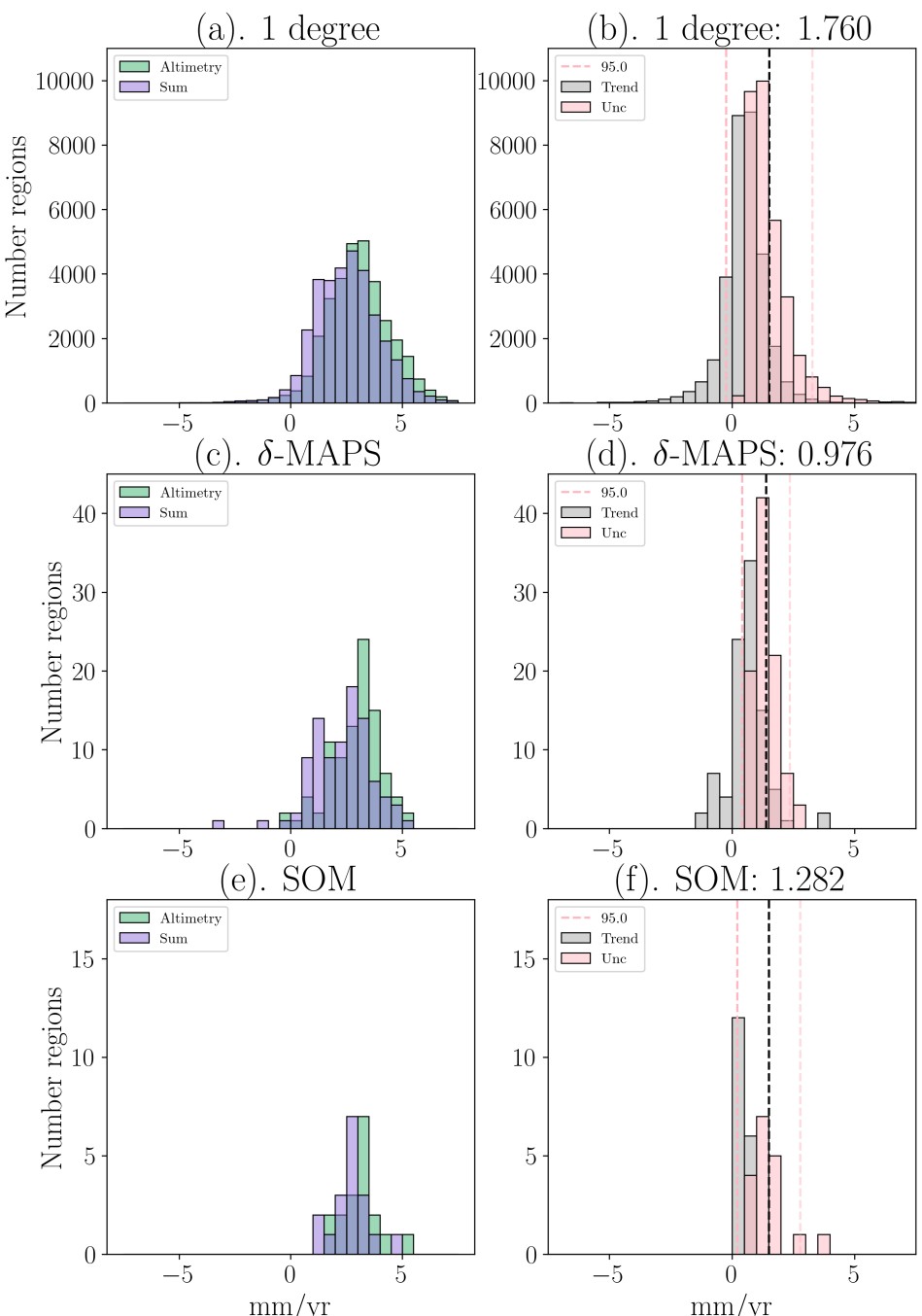

**Figure B2.** Distribution histograms of the altimetry (left column, green blocks), sum of the components (left column, purple blocks) and residuals trend (right column, gray) and uncertainty (right columns, pink), for the 1x1 degree budget (top row), $\delta$-MAPS domains (middle) and SOM domains (bottom). The dashed pink lines indicate the 95% confidence interval of the residuals uncertainty, with the interval width reported in the subplots titles, and was used as a reference for the residuals scatters in Figure 3

**Table B1.** Names of SOM and $\delta$-MAPS domains

| SOM | $\delta$-MAPS | Domain name |
| --- | --- | --- |
| 1 | 82 | Gulf Stream |
| 2 | 24 | Southeast Atlantic |
| 3 | 69 | Malvinas Current |
| 4 | 39 | Central North Atlantic Gyre |
| 5 | 34 | East Africa Atlantic coast |
| 6 | 75 | East Equatorial Atlantic |
| 7 | 63 | Brazil Current |
| 8 | 66 | Northwest European Shelf |
| 9 | 33 | South of Greenland |
| 10 | 88 | Kuroshio Extension |
| 11 | 90 | Northwest Indian Ocean |
| 12 | 45 | ENSO-tongue' |
| 13 | 92 | Southwest Autralia, Freemantle region |
| 14 | 67 | Southweast Indian Ocean |
| 15 | 62 | Southwest Tropical Atlantic Ocean |
| 16 | 89 | West Tropical Pacific Ocean, Australiasian Seas |
| 17 | 77 | Agulhas Current |
| 18 | 85 | Central North Pacific Ocean |

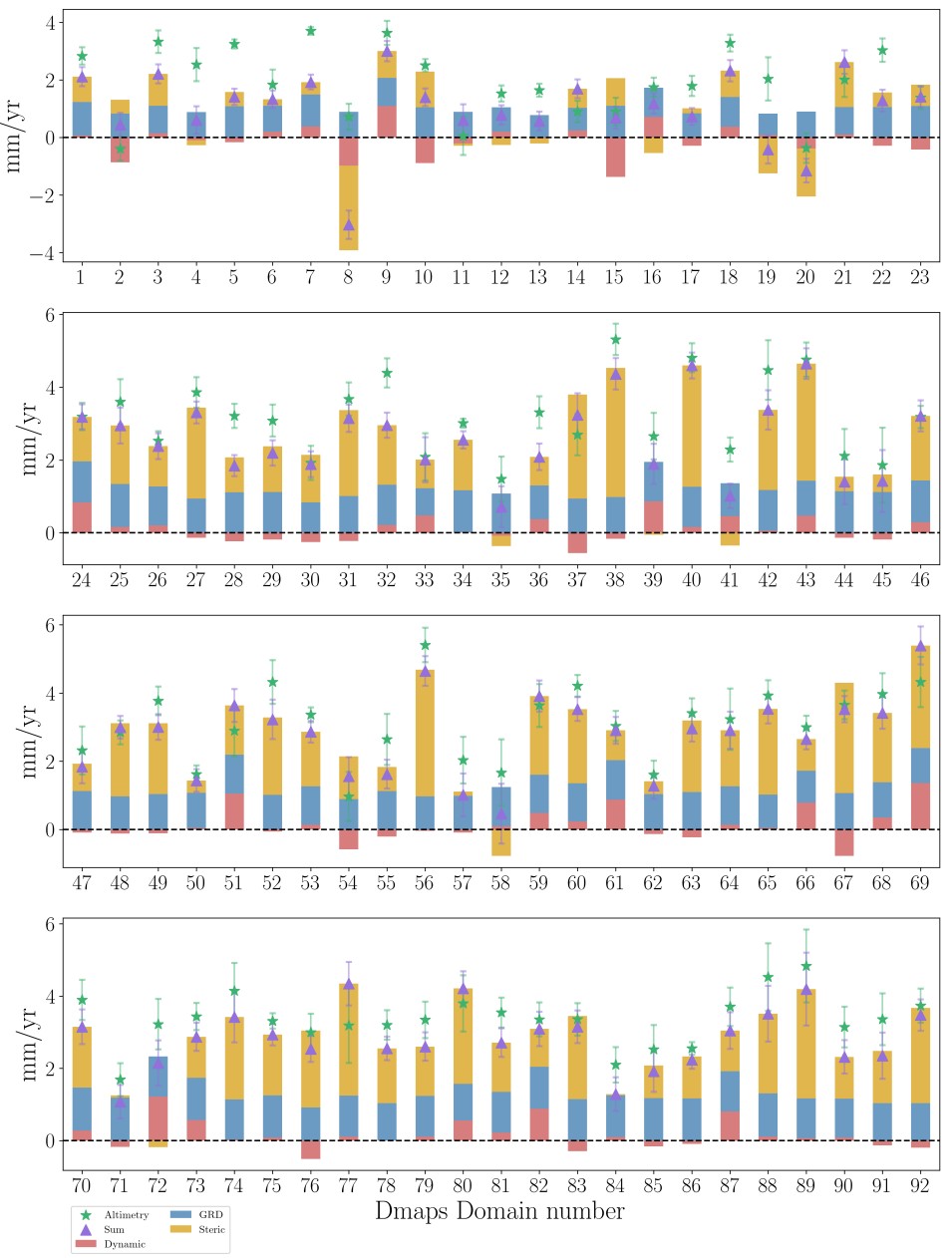

**Figure B3.** Extension of Figure 4, showing the trend contribution for each δ-MAPS domains. An interactive budget map is available at https://carocamargo.github.io/resources/regional-SLB-domains/

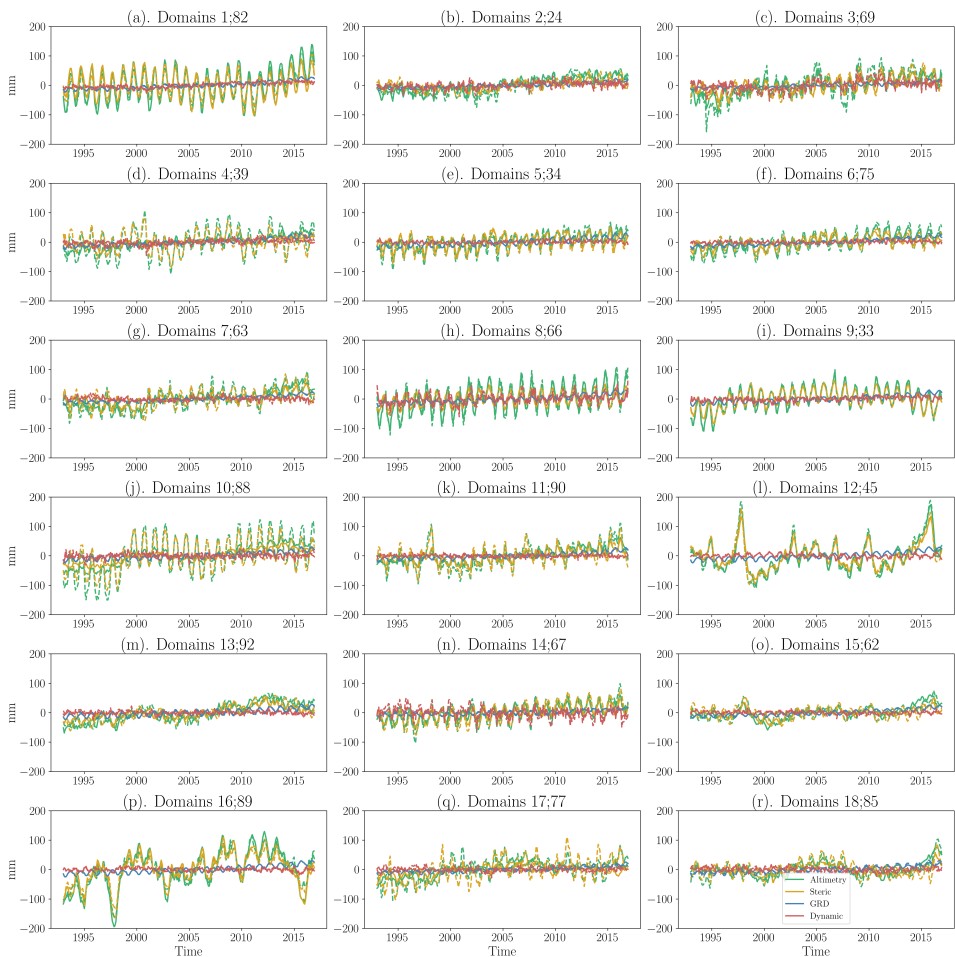

**Figure B4.** Extension of Figure 4, showing the time series for each of the SOM domains and corresponding δ-MAPS domains.

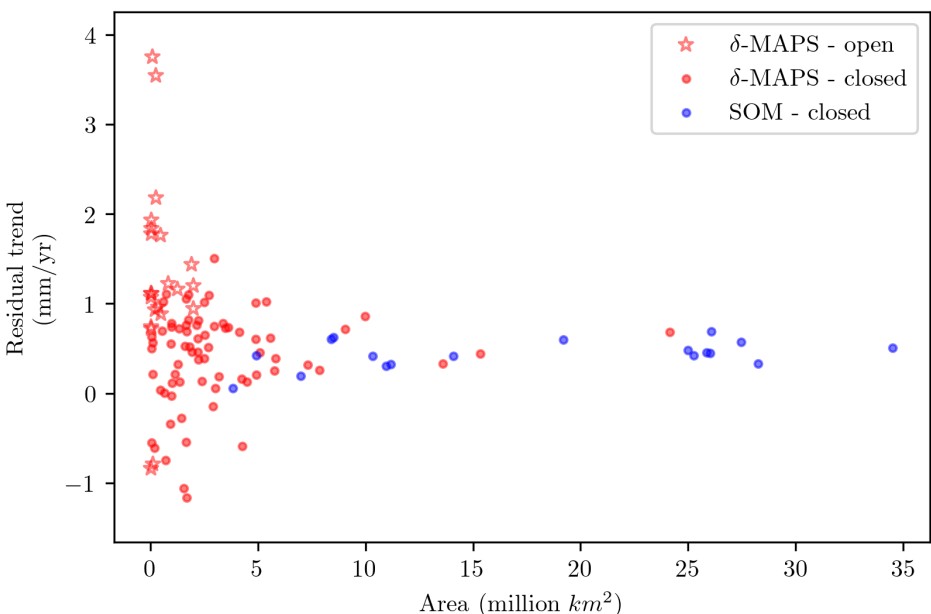

**Figure B5.** Scatter plot of the budget residuals (i.e., altimetry minus sum of components) against the area of each domain for $\delta$-maps (red) and SOM (blue). Stars and circles indicate domains in which the sea-level budget is open and closed, respectively. As the domain area increases, the residuals converge towards 0. Not only all the SOM residuals are within $\pm$1mm/yr, but also 74.2% of the $\delta$-MAPS domains.

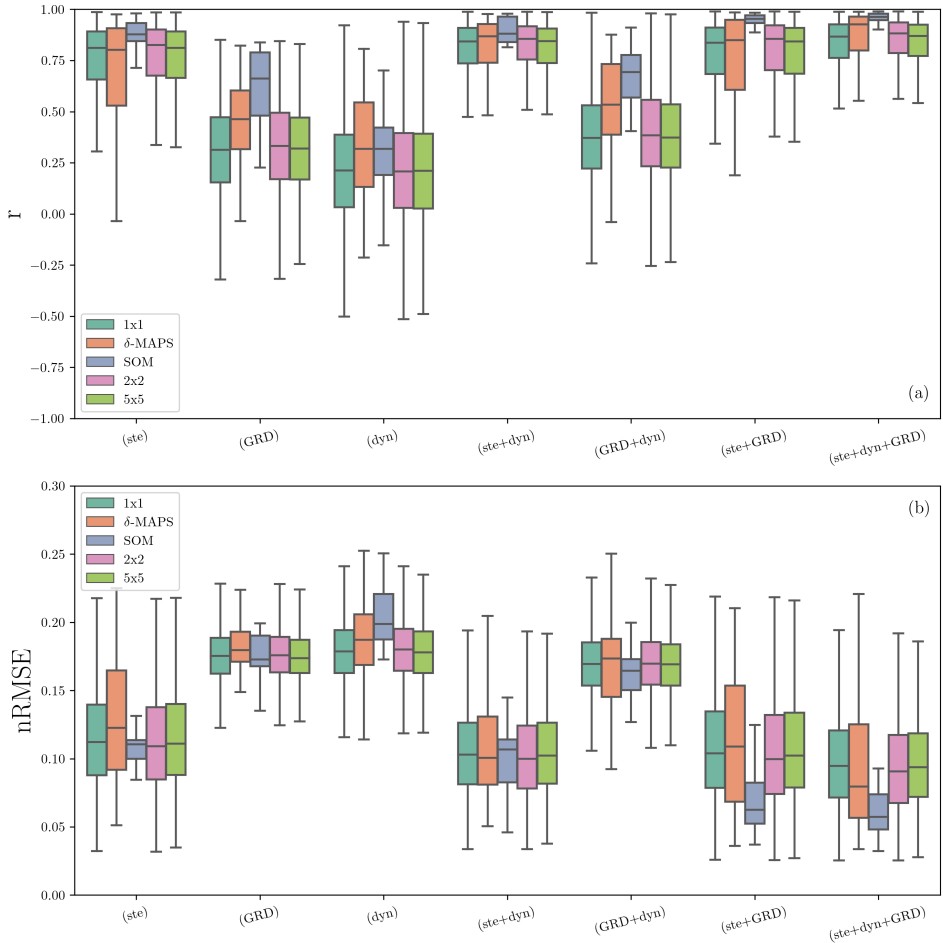

**Figure B6.** Same as Figure 5, but including sea-level budget considering blocks of 2x2 and 5x5 degrees. There is no clear improvement from the 1x1 degree budget to the 2x2 and 5x5, showing the added value of using the $\delta$-MAPS and SOM domains.

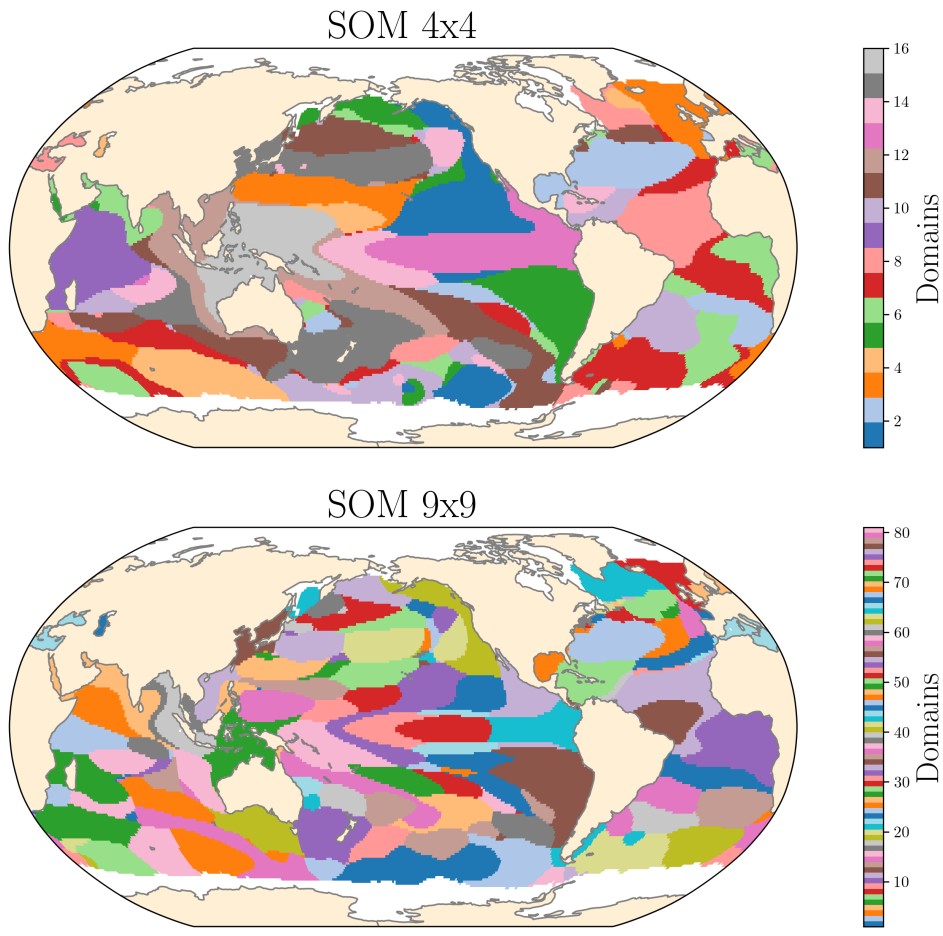

**Figure B7.** SOM domains for a neural map of (a) 4x4 (a) and of (b) 9x9, using the entire ocean as input for the clustering. Note that, even with a larger neural map, the SOM patterns are still different than the $\delta$-MAPS domains (Figure 2), proving that the difference between the extracted patterns are not just a function of the number of SOM neurons, but due to differences between the two methods. It does, however, leads to fewer regions that are geographically distant being clustered in the same domain.

*Author contributions.* CC performed the research and drafted the article. CC, RR and AS designed the study. Self-organizing maps methods and analysis were applied by CC, IH-C and MM. CC and ES performed $\delta$-MAPS analysis. All authors contributed to the interpretation of the results and the writing of the manuscript.

*Competing interests.* The authors declare no competing interests.

*Acknowledgements.* This research was funded by the Netherlands Space Office User Support program (grant no. ALWGO.2017.002).

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
