# Peer review of "Regionalizing the Sea-level Budget With Machine Learning Techniques"

_EGUsphere, 2022_

## Referee Comment (RC1)

**Review of "Regionalizing the Sea-level Budget With Machine Learning Techniques" by Camargo *et al* (2022) for Ocean Science**

Review by Dr Sam Royston, University of Bristol

**Summary**

This work discusses the sea-level budget from observations at different spatial scales. While the global-mean sea-level budget closes for the period of observations from 1993—2016, there remain differences in the budget at smaller scales, which are important to understand. The approach here is to use two unsupervised machine learning methods to define smaller, sub-ocean-basin scale regions with covarying sea level and test the sea-level trend budget at a range of scales. It is an important topic and the paper is well written and clear. I do have a reservation about the use of reanalyses data that dominate both the steric (pre-Argo) and manometric DSL components but the authors have provided sensitivity analysis and comparison with observations (GRACE) that imply the main conclusions of the paper are still valid. There are a few general remarks that I feel the authors should address before publication, but overall it is a substantive piece of work of excellent quality and worthy of publication.

**General Remarks**

The authors do note in Appendix B that there is some circularity in their sea-level (trend) budget. They use reanalysis data for the manometric dynamic SLA and in the steric SLA ensemble, but these reanalyses mostly assimilate altimetric SSHA data; which the authors are then comparing against. This issue is most concerning pre-Argo, as the steric SLA ensemble becomes heavily weighted to the reanalyses products. I don't feel it appropriate to ask the authors to repeat the trends and analysis for the Argo + GRACE period (since ~2005). The authors have done some work to investigate the difference between GRACE observations and the reanalyses DSL (Fig A1) and to sensitivity test using different data sets, including Argo-only steric data sets (Section 4.3). I would like to see the authors move the comment from the Appendix B into the main text, in Section 2.1, with a specific reference to the sensitivity tests they do and the period of data they are choosing to use (for the main result and if some of the sensitivity tests are applied over shorter durations).

The abstract could be more precise, in particular to quantify how 'well' the sea-level trend budget matches. Line 8 says the authors can close the SL(T)B on [some] scales but then line 11 says some regions the SL(T)B does not close – it might be clearer to state that the SLTB closes in 100% of the 18 sub-basin regions defined using SOM, but on smaller scales the SLTB can fail to close.

The authors appear to discuss both a SLB (time series) and SLTB and the manuscript would benefit from clarity between these two metrics. (Line 111 states your SLB is actually a SLTB but Figures 4,5 suggest you are also comparing time series.)

Terminology. Reading the Appendix A is a bit confusing! The authors should clarify if I understand correctly, and perhaps change the terminology if needed for clarity. My understanding is the authors are replicating absolute SSHA (observed with IB-corrected altimetry) with a sum of what they call "steric SLA", "GRD" and a manometric "dynamic SLA" term. The GRD term here, is a combination of the GRD terms applicable to absolute sea-level and the barystatic SLA (i.e. it includes the global-mean manometric change at each time step). I appreciate the citation but I think it could be clearer exactly what this 'GRD' includes. And the "dynamic SLA" term in the main paper is the residual of modelled sterodynamic SLA with the steric contribution removed (assuming IB-corrected is consistent with the Gregory et al, 2019, definition). If this is correct then I think the authors should simplify the description in Appendix A.

**Review of "Regionalizing the Sea-level Budget With Machine Learning Techniques" by Camargo *et al* (2022) for Ocean Science**

Not a necessity, but it would be easier to digest some of the text as Tables or Figures. In Section 2.1 you could tabulate the data sources for each component, description, temporal and spatial resolution, and citations. In Section 4.1, you state that the residuals are improved with scale. I would like to see a scatter plot of the residual (altimetry – sum of components) for the \delta-maps and SOM (i.e. from Fig. 3 c,e) against the area that each of those regions cover to see if there is a simple relationship with scale.

**Line by Line Comments**

Figure titles: What are the error bars shown (1 or 2 standard errors or standard deviations?).

Abstract Line 7: "besides indicating" can be simplified to "indicate" ("The extracted domains provide … and indicate …")

Abstract Line 8: Suggest you be specific as most readers will skim the abstract. State what period you can close the observational sea level budget for. State within what error it closes (1 or 2 standard errors?). Do your time series SLB also close within 1 or 2 standard errors or is it the trends.

Abstract Line 9: Suggest replace "transport" with "exchange".

Lines 40-45: Novi et al (2021) mostly discuss SST so citation should be moved to preceding sentence.

Lines 68-71 etc: What temporal resolution are you using for steric, DSL, GRD and total SSH (altimetry)?

Line 80 etc: Can you state the temporal and spatial resolution of the Camargo (2020) ensemble steric SL data set, the Camargo et al (2022) GRD data set and the ensemble DSL reanalyses?

Line 101: Can you clarify, do you mean you take the steric SLA from the total SLA for each model in turn.

Line 119: How do you combine 'uncertainty' in the spatial averages in the SOM / delta-map regions?

Section 2.3: Sorry if I missed it, but it isn't clear which data set(s) you use for the clustering analysis (Satellite altimetry $\eta_{total}$ or the reanalysis products you also use; what temporal extent and sampling of the data is used). This is particularly important when you discuss possible mechanisms for the cross-correlation, in Section 3.

Line 135: Do you mean a Gaussian filter with 300 km **half-**width (i.e. the power is 50% at 300 km radial distance), akin to GRACE resolution?

Lines to 172: Out of interest what value of \delta did you use for the \delta-maps threshold? i.e. what is the Figure 3 "uncertainty", is it 1-sigma standard error?

Section 3: This is an interesting discussion but it would be worth noting that the SOM and \delta-maps don't account for auto-correlation in time, i.e. the time lag in the progression of a signal across the ocean basin. So signals that are rapidly propagating compared with the time-sample of your data (monthly?) correlate – typically barotropic, manometric signals - but slower-propagating signals such as the first baroclinic mode will lose correlation in space. So baroclinic signals near the equator, which can propagate faster than those at high latitudes, appear 'better' correlated. i.e. the temporal sampling of the observations that you provide these algorithms dictates which processes appear 'coherent'.

**Review of "Regionalizing the Sea-level Budget With Machine Learning Techniques" by Camargo *et al* (2022) for Ocean Science**

Figure 4 c,d: There is a lot going on there, you need to label solid and dashed lines or just present the time series for one ML region set.

Section 4.3 is very useful to see the sensitivity of the SL(T)B to the size and clustering of regions and the data sets used, but omitting components from the SLB isn't really a sensitivity? Fig 5 relating to the components isn't that informative (sorry!) because the dominance of the steric signal relates to your choice of sampling / data (monthly data that retains the seasonal cycle). But the comparison of the different domain scales is informative. (Actually the box plots in Fig 5a,b ste+dyn+GRD show the 'improvement' for the larger SOM maps that I thought would be interesting to scatter plot with area.)

Line 299: As the authors have shown in Figs 4d,e the seasonal cycle is predominantly steric, which gives rise to a 'better' correlation and lower RMSE since most of the variance in monthly SSHA is the seasonal signal. Line 299, to reduce the apparent dominance of the steric signal in your analysis, you would need to deseason steric SLA and SSHA at each location (not just remove a global mean at each time step).

Line 303: Additionally the more samples you average, the smaller the standard error.

Line 306: Just a note, that measurement errors between altimetry and the sum of components "average out" to zero only if the errors are uncorrelated in space, i.e. they are random, at the scale you are averaging over.

Lines 319-324 and Fig 6: This is a really useful analysis and allays concerns about the data choice that detract from your main conclusions.

Conclusion: You could also add that coherent total sea surface height change might not be same coherency as component parts (steric, manometric dynamic and GRD), so depending it could be 'better' to isolate manometric-dominated variability from steric dominated variability first and then cluster them separately.

Appendix A: Line 377: The "as a result of steric changes" is confusing here and I think unnecessary. I think what the authors are doing is replicating absolute SSHA with a sum of steric SLA, GRD (which here the authors define to include barystatic SLC) and a manometric dynamic SLA term. The latter is, in line with the Gregory et al (2019) definition, the residual of modelled sterodynamic SLA with the steric contribution removed. So it includes mass exchange at any point; changes to the ocean circulation and atmospheric redistribution effects (by mass redistribution, by wind stress and by non-linear interaction due to density changes). (And if the model output weren't IB-corrected, the local atmospheric pressure changes.)

Appendix A: Lines 409-411: There is a strong difference between GRACE and ocean reanalysis in the South Atlantic and you are using a period with a good coverage of Argo float and ship-based in-situ data to characterise the steric component in the reanalysis. You conclude that a source of dynamic SLC may be poorly parameterised in the reanalyses, and perhaps the difference between different reanalyses on different resolutions could be interesting. But my counter-argument would be that the difference covers a large spatial area pointing towards the GRACE spherical harmonic solutions and low-degree corrections.

---

## Author Comment (AC2)

**Authors' Response to Reviewer 1**

> **General Comment.** This work discusses the sea-level budget from observations at different spatial scales. While the global-mean sea-level budget closes for the period of observations from 1993—2016, there remain differences in the budget at smaller scales, which are important to understand. The approach here is to use two unsupervised machine learning methods to define smaller, sub-ocean-basin scale regions with covarying sea level and test the sea-level trend budget at a range of scales. It is an important topic and the paper is well written and clear. I do have a reservation about the use of reanalyses data that dominate both the steric (pre-Argo) and manometric DSL components but the authors have provided sensitivity analysis and comparison with observations (GRACE) that imply the main conclusions of the paper are still valid. There are a few general remarks that I feel the authors should address before publication, but overall it is a substantive piece of work of excellent quality and worthy of publication.

**Response:**

Dear Dr. Royston,

Thank you for your feedback and positive review. We understand your reservation, and hope that the answers to your comments have clarified the issues. We have carefully addressed all the issues item by item as follows.

Kind regards,

Carolina Camargo, on behalf of the authors

**Comment 1**

The authors do note in Appendix B that there is some circularity in their sea-level (trend) budget. They use reanalysis data for the manometric dynamic SLA and in the steric SLA ensemble, but these reanalyses mostly assimilate altimetric SSHA data; which the authors are then comparing against. This issue is most concerning pre-Argo, as the steric SLA ensemble becomes heavily weighted to the reanalyses products. I don't feel it appropriate to ask the authors to repeat the trends and analysis for the Argo + GRACE period (since 2005). The authors have done some work to investigate the difference between GRACE observations and the reanalyses DSL (Fig A1) and to sensitivity test using different data sets, including Argo-only steric data sets (Section 4.3). I would like to see the authors move the comment from the Appendix B into the main text, in Section 2.1, with a specific reference to the sensitivity tests they do and the period of data they are choosing to use (for the main result and if some of the sensitivity tests are applied over shorter durations)

**Response:** Indeed, pior to Argo, the steric datasets rely more on reanalyses products, but note that the our ensemble steric estimate is composed of 15 data sets: 5 based on Argo, 5 based on multiple in-situ observations, and 5 based on ocean reanalyses. So while prior to 2002 the ensemble relies more on ocean reanalysis, it does not become heavily weighted by it, as we still have 5 other in-situ data sets. Nonetheless, there is some circularity in using ocean reanalysis' sea-surface height, which incorporates satellite altimetry, to estimate the dynamic component, and then use this dynamic estimate to compare with satellite altimetry. We have moved the comment about the circularity and the validation with GRACE to the main text:

> $\eta_{DSL}$ is computed from the sea-surface height of 5 ocean reanalyses (Table **??**), by first removing the time-varying global mean from the sea-surface height, and then by removing the local steric anomaly. This procedure is done in each ocean

reanalysis individually, and we then combine the 5 estimates into an ensemble. We acknowledge that this method introduces some circularity to the budget analysis: the reanalysis, used to obtain $\eta_{DSL}$, assimilate satellite sea-surface height, and in the budget analysis we compare this estimate with satellite sea-surface height ($\eta_{total}$). Compared with the $\eta_{DSL}$ estimated from Gravity Recovery and Climate Experiment Satellite (GRACE, Tapley et al., 2004), $\eta_{DSL}$ sea-level trends from 2005-2015 agree on large scale patterns and magnitude of dynamic changes (Figure A1). Note that our budget components do not incorporate GRACE mass changes over the oceans, hence it is an independent estimate for validation.

**Comment 2**

The abstract could be more precise, in particular to quantify how 'well' the sea-level trend budget matches. Line 8 says the authors can close the SL(T)B on [some] scales but then line 11 says some regions the SL(T)B does not close – it might be clearer to state that the SLTB closes in 100% of the 18 sub-basin regions defined using SOM, but on smaller scales the SLTB can fail to close.

**Response:** We have modified the abstract to be more precise:

Using these domains we can close, within 1-sigma uncertainty, the sub-basin regional sea-level budget from 1993-2016 in 100% and 83% of the SOM and $\delta$-MAPS regions, respectively.

> **Comment 3**
>
> The authors appear to discuss both a SLB (time series) and SLTB and the manuscript would benefit from clarity between these two metrics. (Line 111 states your SLB is actually a SLTB but Figures 4,5 suggest you are also comparing time series.)

**Response:** Thank you for your comment, which we agree. We modified line 111 to clarify that the budget analysis included both time series and trends:

> Our sea-level budget includes the comparison of sea-level time series, trends and associated uncertainties.

We also modified the title of section 4.1 to clearly state that we discuss trends in that section.

> **Comment 4**
>
> Terminology. Reading the Appendix A is a bit confusing! The authors should clarify if I understand correctly, and perhaps change the terminology if needed for clarity. My understanding is the authors are replicating absolute SSHA (observed with IB-corrected altimetry) with a sum of what they call "steric SLA", "GRD" and a manometric "dynamic SLA" term. The GRD term here, is a combination of the GRD terms applicable to absolute sea-level and the barystatic SLA (i.e. it includes the global- mean manometric change at each time step). I appreciate the citation but I think it could be clearer exactly what this 'GRD' includes. And the "dynamic SLA" term in the main paper is the residual of modelled sterodynamic SLA with the steric contribution removed (assuming IB-corrected is consistent with the Gregory et al, 2019, definition). If this is correct then I think the authors should simplify the description in Appendix A.

**Response:** That is correct: we compare the absolute SSHA from altimetry with the sum of steric, GRD and (manometric-)dynamic SLA. We modified Appendix A to be clearer:

> GRACE measures total mass changes, which can be used to derive estimates of manometric sea-level change over the oceans, that is the sea-level change in response to both the dynamic ocean mass redistribution ($\eta_{DSL}$) and to mass redistribution due to the land-ocean mass exchange ($\eta_{GRD}$) (Chambers et al., 2004; Royston et al., 2020).

And we clarified it in the main text:

> The dynamic component ($\eta_{DSL}$, Figure 1e) refers to mass changes driven by bottom pressure changes, that is, the redistribution of mass that was already in the oceans. Note that, by our definition, the dynamic sea-level change ($\eta_{DSL}$) is part of the ocean dynamic sea-level change ($\Delta_\zeta$, Gregory et al., 2019), the latter also including the effect of local steric anomalies ($\eta'_{SSL}$). That is, the dynamic term here is the residual of the sterodynamic sea-level change with the steric contribution removed (Gregory et al., 2019)
* * *
**Comment 5**

Not a necessity, but it would be easier to digest some of the text as Tables or Figures. In Section 2.1 you could tabulate the data sources for each component, description, temporal and spatial resolution, and citations. In Section 4.1, you state that the residuals are improved with scale. I would like to see a scatter plot of the residual (altimetry – sum of components) for the $\delta$-maps and SOM (i.e. from Fig. 3 c,e) against the area that each of those regions cover to see if there is a simple relationship with scale.

**Response:** Thank you for the suggestion. We have added a table at the end of Section 2.1, summarizing the components description, temporal and spatial resolution and data sources:

dynamics. A summary of the budget components and data sets sources is given on Table 1. Note that all the used data sets have monthly temporal resolution and a 1°x1° spatial resolution.

**Table 1.** Summary of the sea-level budget components and data sources used in this manuscript.

| Symbol | Name | Description | Reference |
|---|---|---|---|
| $\eta_{total}$ | Observed change | Total sea-level change from satellite altimetry | Ensemble of CMEMS (CMEMS, 2022), JPL MEaSUREs (Zlotnicki et al., 2019), SLcci (SLcci, 2022) and CSIRO (CSIRO, 2022) |
| $\eta_{SSL}$ | Steric expansion | Full depth density-driven sea-level change due to ocean temperature and salinity variations | Camargo et al. (2020) and Purkey and Johnson (2010) |
| $\eta_{GRD}$ | Mass change | Contemporary ocean mass redistribution due to the land-ocean mass exchange | Camargo et al. (2022) |
| $\eta_{DSL}$ | Dynamic change | Mass redistribution due to purely ocean dynamics | Ensemble of SODA (Carton et al., 2018), C-GLORS (Storto and Masina, 2016), GLORYS (Garric and Parent, 2017), FOAM-GloSea (Blockley et al., 2014; Maclachlan et al., 2015) and ORAS (Zuo et al., 2019) |

We have also made a scatter plot of the residual against the area of each region (Figure 1). This figure shows how the residual values are much larger when smaller regions are considered, confirming that the residuals have improved with scale. We have added it to Appendix B, as it illustrates the conclusions drawn in Section 4.

[Figure]

Figure 1: Scatter plot of the residuals (i.e., altimetry minus sum of components) against region size for $\delta$-maps (red) and SOM (blue).

**Line by Line Comments**

**Comment 1**

Figure titles: What are the error bars shown (1 or 2 standard errors or standard deviations?).

**Response:** The error bars show the 1-sigma uncertainty. We have added this information to the caption of Figure 3 and 4. On Figure 5, the whiskers (not error bars) show the full distribution, while the box shows the quantiles of the data.

**Comment 2**

Abstract Line 7: "besides indicating" can be simplified to "indicate" ("The extracted domains provide ... and indicate ...")

**Response:** We have modified it accordingly.

> **Comment 3**
>
> Abstract Line 8: Suggest you be specific as most readers will skim the abstract. State what period you can close the observational sea level budget for. State within what error it closes (1 or 2 standard errors?). Do your time series SLB also close within 1 or 2 standard errors or is it the trends.

**Response:** We have modified the abstract to be more specific.

> Using these domains we can close, within 1-sigma uncertainty, the sub-basin regional sea-level budget from 1993-2016 in 100% and 83% of the SOM and $\delta$-MAPS regions, respectively.

> **Comment 4**
>
> Abstract Line 9: Suggest replace "transport" with "exchange".

**Response:** We have modified it accordingly.

> **Comment 5**
>
> Lines 40-45: Novi et al (2021) mostly discuss SST so citation should be moved to preceding sentence.

**Response:** We moved the citation to the preceding sentence, as suggested.

**Comment 6**

Lines 68-71 etc: What temporal resolution are you using for steric, DSL, GRD and total SSH (altimetry)?

**Response:** We used a monthly temporal resolution for all estimates. We now clarify this when presenting the satellite altimetry data sets and the other contributions:

> We use multi-mission gridded Level-4 data from 4 distribution centers: CMEMS (CMEMS, 2022), JPL MEaSUREs (Zlotnicki et al., 2019), SLcci (SLcci, 2022) and CSIRO (CSIRO, 2022). All of these products use the same reference ellipsoid model (GRS80/WGS). All data sets have a monthly temporal resolution, except for JPL MEaSUREs time series which provides sea surface height data every 5 days and was averaged into monthly means. All data is regridded to 1x1 degree, selected within 66°S to 66°N of latitude, and combined into an ensemble mean, to avoid systematic errors. All the following data sets have the same spatio-temporal characteristics: monthly mean values on a 1°x1° map.

**Comment 7**

Line 80 etc: Can you state the temporal and spatial resolution of the Camargo (2020) ensemble steric SL data set, the Camargo et al (2022) GRD data set and the ensemble DSL reanalyses?

**Response:** Following the previous reply, we have added this information to the text when the first data sets are introduced. This information is also reinforced when table 1 is introduced (see reply to General Comment 5).

**Comment 8**

Line 101: Can you clarify, do you mean you take the steric SLA from the total SLA for each model in turn.

**Response:** We do it model by model, and then combine the 5 estimates into an ensemble. We have clarified the sentence accordingly:

> $\eta_{DSL}$ is computed from the sea-surface height of 5 ocean reanalyses (Table 1), by first removing the time-varying global mean from the sea-surface height, and then by removing the local steric anomaly. This procedure is done in each ocean reanalysis individually, and we then combine the 5 estimates into an ensemble.

**Comment 9**

Line 119: How do you combine 'uncertainty' in the spatial averages in the SOM / delta-map regions?

**Response:** We took the area-weighted value of the uncertainties within a SOM/delta-map region. We have added this information to the main text to have it clear for the readers

> For each SOM and $\delta$-MAPS region we take the area-weighted spatial average of the time series, trend and uncertainties.

**Comment 10**

Section 2.3: Sorry if I missed it, but it isn't clear which data set(s) you use for the clustering analysis (Satellite altimetry $\eta_{total}$ or the reanalysis products you also use; what temporal extent and sampling of the data is used). This is particularly important when you discuss possible mechanisms for the cross-correlation, in Section 3.

**Response:** For the clustering analysis we used satellite altimetry from 1993 to 2019 ($\eta_{total}$), as stated in Line 132. We added now the information about the temporal and spatial resolution:

> For both clustering techniques we use 1°x1° monthly satellite altimetry time-series (CMEMS, 2022), for 1993-2019, as input.

**Comment 11**

Line 135: Do you mean a Gaussian filter with 300 km half-width (i.e. the power is 50% at 300 km radial distance), akin to GRACE resolution?

**Response:** Yes, that is what was meant. We modified it to 'half-width'.

**Comment 12**

Lines to 172: Out of interest what value of $\delta$ did you use for the $\delta$-maps threshold? i.e. what is the Figure 3 "uncertainty", is it 1-sigma standard error?

**Response:** The $\delta$ value used in $\delta$-MAPS is not related to the uncertainty shown in Figure 3. In Figure 3 we show the 1-sigma uncertainty, as mentioned in Section 2.2 (line 113) (see also reply to Line-by-line Comment 1). For $\delta$-maps, we actually do not

choose the $\delta$ parameter but the $\alpha$ parameter, which defines the significance level of the homogeneity test between grid cells. We set $\alpha$ to 0.01, meaning that every cell which is included in a domain has a similarity with the other cells of 99% significance level.
* * *
**Comment 13**

Section 3: This is an interesting discussion but it would be worth noting that the SOM and $\delta$-maps don't account for auto-correlation in time, i.e. the time lag in the progression of a signal across the ocean basin. So signals that are rapidly propagating compared with the time-sample of your data (monthly?) correlate – typically barotropic, manometric signals - but slower-propagating signals such as the first baroclinic mode will lose correlation in space. So baroclinic signals near the equator, which can propagate faster than those at high latitudes, appear 'better' correlated. i.e. the temporal sampling of the observations that you provide these algorithms dictates which processes appear 'coherent'.
* * *
**Response:** Thank you for this interesting comment. We think this is a relevant information to be added to the the discussion. Indeed, the time resolution of data used in this study is not appropriate to adequately identify the propagation of fast signals. However, using SOM one can infer dynamical propagation of signals, as long as the input data has a high frequency temporal resolution. For instance in Liu et al. (2016), it was explored the penetration of the Loop Current into the Gulf of Mexico through the combined SOM analysis in the time and space domain with wavelets, based on daily sea-surface height data. That is, you can analyze how the identified SOM regions correlate in time, for instance computing the cross correlation (or cross-wavelets) between the temporal patterns. We have added this discussion to the first paragraph of Section 3 as follows:

It is also important to note that these clustering methods do not account for auto-correlation in time, that is the time lag in the progression of a signal across the ocean basin. Since we use monthly data, signals that propagate faster than a month (typically barotropic) will be more clearly correlated in our clustering. On the other hand, slower propagating signals, such as the first baroclinic mode, will lose correlation in space, and will not be represented in the identified domains.

**Comment 14**

Figure 4 c,d: There is a lot going on there, you need to label solid and dashed lines or just present the time series for one ML region set.

**Response:** Thank you for pointing this out. We chose to reformat the figure, in a way that panels c and d take the entire width of the figure. Since the color of the dashed lines represents the same as the solid lines, we chose to just add the difference between the solid and dashed lines in the caption.

**Comment 15**

Section 4.3 is very useful to see the sensitivity of the SL(T)B to the size and clustering of regions and the data sets used, but omitting components from the SLB isn't really a sensitivity? Fig 5 relating to the components isn't that informative (sorry!) because the dominance of the steric signal relates to your choice of sampling / data (monthly data that retains the seasonal cycle). But the comparison of the different domain scales is informative. (Actually the box plots in Fig 5a,b ste+dyn+GRD show the 'improvement' for the larger SOM maps that I thought would be interesting to scatter plot with area.)

**Response:** We agree that omitting components of the SLB is not a real sensitivity analysis, but it does show how the budget improves as we include more components (as expected), and highlights the importance of including both the deep ocean and the dynamic component in the budget. Thus, we decided to keep Figure 5 how it is, but we changed the title of Section 4.3 to "Sea-level Budget Performance", and do not say we are doing a sensitivity analysis, but instead just investigating how the closure of the budget changes when considering different components, area, and datasets:

> ## 4.3. Sea-Level Budget Performance
>
> Here, we investigate the closure of the budget considering (i) the components included in the budget, (ii) the size of the domains and the clustering method, and (iii) the data sets used for each component.

However, it is true that the correlation and RMSE will improve with the addition of the steric contribution, because of its dominance on the seasonal cycle of the altimetry time series. We added this information to Line 298:

> While we get a poorer performance when only considering the dynamic or the GRD component, the budget with only steric already performs relatively well. The improved correlation and lower RMSE with the steric component is probably a result of the seasonal cycle being predominantly steric.

**Comment 16**

Line 299: As the authors have shown in Figs 4d,e the seasonal cycle is predominantly steric, which gives rise to a 'better' correlation and lower RMSE since most of the variance in monthly SSHA is the seasonal signal. Line 299, to reduce the apparent dominance of the steric signal in your analysis, you would need to deseason steric SLA and SSHA at each location (not just remove a global mean at each time step).

**Response:** Indeed, not only removing the global mean but also removing the seasonal cycle would be important to reduce the dominance of the steric signal. We have, however, removed this sentence from the manuscript.

**Comment 17**

Line 303: Additionally the more samples you average, the smaller the standard error.

**Response:** Thank you for highlight this point. Indeed, the more samples are averaged, the smaller the error. We added this information to the manuscript.

**Comment 18**

Line 306: Just a note, that measurement errors between altimetry and the sum of components "average out" to zero only if the errors are uncorrelated in space, i.e. they are random, at the scale you are averaging over.

**Response:** Thank you for the comment. We added this note together with the previous comment:

Additionally the more samples are averaged the smaller the standard error. However, the measurement errors between altimetry and the sum of components will only compensate each other if the errors are uncorrelated in space, i.e., if they are random at the scale being analyzed.

And indeed, as mentioned in General Comment 5, the scatter plot of the residuals with the domain area confirms the improvement of the budget performance as shown in Figure 5. We have added the scatter plot to Appendix B.

**Comment 19**

Lines 319-324 and Fig 6: This is a really useful analysis and allays concerns about the data choice that detract from your main conclusions.

**Response:** Thank you.

**Comment 20**

Conclusion: You could also add that coherent total sea surface height change might not be same coherency as component parts (steric, manometric dynamic and GRD), so depending it could be 'better' to isolate manometric-dominated variability from steric dominated variability first and then cluster them separately.

**Response:** Thank you for your suggestion. That is indeed the case, and we tested SOM on different budget components, and it does give different patterns, specially when GRD patterns are used as input. We now comment on this in the second paragraph of the conclusions:

... we applied a neural network approach, SOM, and a deep-network detection method, $\delta$-MAPS, to identify domains of coherent sea-level variability (Figure 2). Note however, that the coherent patterns will be different whether total sea surface height or the individual components (steric, dynamic, GRD) are considered. Hence, depending on the purpose of the study, one should first remove the unwanted components from total sea-surface height, and then perform the clustering.
* * *
**Comment 21**

Appendix A: Line 377: The "as a result of steric changes" is confusing here and I think unnecessary. I think what the authors are doing is replicating absolute SSHA with a sum of steric SLA, GRD (which here the authors define to include barystatic SLC) and a manometric dynamic SLA term. The latter is, in line with the Gregory et al (2019) definition, the residual of modelled sterodynamic SLA with the steric contribution removed. So it includes mass exchange at any point; changes to the ocean circulation and atmospheric redistribution effects (by mass redistribution, by wind stress and by non-linear interaction due to density changes). (And if the model output weren't IB-corrected, the local atmospheric pressure changes.)

**Response:** The "as a result of steric changes" had been added as a complement only to the last part of that sentence (ocean bottom pressure changes), to differentiate from the GRD-barystatic sea-level change, which is also a response to ocean bottom pressure changes. But we can see how this can be misinterpreted. Hence, we modified the definition as suggested:

The dynamic redistribution of mass due to ocean circulation and atmospheric redistribution effects is known as dynamic sea-level change ($\eta_{DSL}$ Gregory et al., 2019; Landerer et al., 2007). $\eta_{DSL}$ refers to mass changes driven by bottom pressure

changes, that is, the redistribution of mass that was already in the oceans, and includes mass exchange at any point by mass redistribution, by wind stress and by non-linear interaction due to density changes.

**Comment 22**

Appendix A: Lines 409-411: There is a strong difference between GRACE and ocean reanalysis in the South Atlantic and you are using a period with a good coverage of Argo float and ship-based in-situ data to characterise the steric component in the reanalysis. You conclude that a source of dynamic SLC may be poorly parameterised in the reanalyses, and perhaps the difference between different reanalyses on different resolutions could be interesting. But my counter-argument would be that the difference covers a large spatial area pointing towards the GRACE spherical harmonic solutions and low-degree corrections.

**Response:** Thank you for pointing this out. Indeed, this difference could also be due to issues in the GRACE harmonic solutions. We have added a comment to the text:

Another strong divergence is seen in the South Atlantic, where the positive trends of GRACE are not represented in the reanalysis, possibly suggesting that a source of dynamic sea-level change is not well parameterized in the reanalysis. Alternatively, this divergence might also be an artefact of the GRACE spherical harmonic solutions and low-degree corrections.

**References**

Chambers, D. P., Wahr, J., & Nerem, R. S. (2004). Preliminary observations of global ocean mass variations with GRACE. *Geophysical Research Letters*, *31*(13), 1–4. https://doi.org/10.1029/2004GL020461

CMEMS. (2022). GLOBAL OCEAN GRIDDED L4 SEA SURFACE HEIGHTS AND DE-RIVED VARIABLES REPROCESSED (1993-ONGOING) SEALEVEL_GLO_PHY_L4_MY_0 https://doi.org/https://doi.org/10.48670/moi-00148

CSIRO. (2022). Combined TOPEX Poseidon, Jason-1, Jason-2 OSTM, Jason-3 near-global gridded monthly-average sea level product. https://doi.org/http://www.cmar.csiro.au/sealevel/sl_data_cmar.html

Gregory, J. M., Griffies, S. M., Hughes, C. W., Lowe, J. A., Church, J. A., Fukimori, I., Gomez, N., Kopp, R. E., Landerer, F., Cozannet, G. L., Ponte, R. M., Stammer, D., Tamisiea, M. E., & van de Wal, R. S. (2019). Concepts and Terminology for Sea Level: Mean, Variability and Change, Both Local and Global. *Surveys in Geophysics*, *40*(6), 1251–1289. https://doi.org/10.1007/s10712-019-09525-z

Landerer, F. W., Jungclaus, J. H., & Marotzke, J. (2007). Ocean bottom pressure changes lead to a decreasing length-of-day in a warming climate. *Geophys. Res. Lett.*, *34*. http://dx.doi.org/10.1029/2006GL029106

Liu, Y., Weisberg, R. H., Vignudelli, S., & Mitchum, G. T. (2016). Patterns of the loop current system and regions of sea surface height variability in the eastern Gulf of Mexico revealed by the self-organizing maps. *Journal of Geophysical Research: Oceans*, *121*(5), 2347–2366. https://doi.org/10.1002/2015JC011493.Received

Royston, S., Vishwakarma, B. D., Westaway, R., Rougier, J., Sha, Z., & Bamber, J. (2020). Can We Resolve the Basin-Scale Sea Level Trend Budget From GRACE Ocean Mass? *Journal of Geophysical Research: Oceans*, *125*(1), 1–16. https://doi.org/10.1029/2019JC015535

SLcci. (2022). Time series of gridded sea level anomalies. https://doi.org/10.5270/esa-sea

Tapley, B. D., Bettadpur, S., Watkins, M., & Reigber, C. (2004). The Gravity Recovery and Climate Experiment: Mission Overview and Early Results. *Geophysical Research Letters*, *31*(L09607), 1–4. https://doi.org/10.1029/2004GL019920

Zlotnicki, V., Qu, Z., Willis, J. K., Ray, R., & Hausman, J. (2019). *JPL MEASURES Gridded Sea Surface Height Anomalies Version JPL1812* (tech. rep.). PO.DAAC. CA, USA. https://doi.org/doi.org/10.5067/SLREF-CDRV2

---

## Author Comment (AC3)

**Authors' Response to Reviewer 2**

**General Comment.** In this manuscript Camargo and colleagues analyze the regional sea level budget (i.e., the sum of individually measured/modelled contributions) to satellite altimetry over the 1993 to 2016 period. They specifically focus on the effect of spatial averaging on the uncertainties in budget closure. For spatial averaging they incorporate an a priori pattern recognition (two different approaches) step, which identifies clusters of homogeneous regions that are then averaged for the budget analysis. They demonstrate that clustering generally improves the budget closure and works significantly better than just using larger blocks. They also demonstrate the importance of the inclusion of an ocean bottom pressure term to the sterodynamic component. Overall, this is a very well written paper using novel approaches with several interesting findings. I therefore have no major reservations regarding the publication of the paper in Ocean Science. Below I provide a couple of comments and suggestions.

**Response:**

Dear Reviewer,

Thank you for your feedback and positive review. We have addressed all the issues item by item as follows.

Kind regards,

Carolina Camargo, on behalf of the authors

> **Comment 1**
>
> I hope I did not overlook anything, but it seems that the authors compare geocentric sea level from satellite altimetry to relative sea level from the budget components, as their budget components also seem to contain crustal components of GRD terms due to contemporary mass change!? To my understanding one must either add those components to satellite altimetry, or only consider the geoid variations in the budget. The term has a substantial contribution to regional sea level according to Frederikse, Riva, et al. (2017)

**Response:** Indeed, altimetry sea level should not be directly compared to relative sea-level change. We understand the confusion of the reviewer, because we did not mention in Section 2.1 that the GRD fingerprints we use represent absolute/geocentric sea-level change. When solving the sea-level equation, both relative and absolute fields are computed. Here we used absolute sea-level change fields. To clarify this important issue, we added the following:

> For the GRD component, we use the estimates from Camargo et al. (2022), which includes the geocentric sea level response to changes on the Antarctic and Greenland ice sheets, glaciers and terrestrial water storage.

**Line by Line Comments**

> **Comment 1**
>
> Line 20: The inverse barometer contribution is missing here

**Response:** Apologies for the confusion. In this sentence we had mentioned only some examples of the processes responsible for the regional differences. But we agree that the

inverse barometer is an important contribution leading to regional differences. Hence, we modified the line accordingly:

> Ocean dynamics, land ice mass changes and associated gravitational effects, vertical land movement and the inverse barometer effect are some of the processes responsible for these regional differences (e.g., Slangen et al., 2017; Stammer et al., 2013).

**Comment 2**

Line 32: or for individual coastline stretches characterized by coherent variability (**Frederikse2016a**; Dangendorf et al., 2021; Frederikse et al., 2016). It has also been closed at a tide gauge level by Wang et al. (2021).

**Response:** References added as follows:

> The sea-level budget has also been analysed for individual coastline stretches characterized by coherent variability (Dangendorf et al., 2021; Frederikse et al., 2016; Frederikse, Simon, et al., 2017; Rietbroek et al., 2016), and at individual tide gauges (Wang et al., 2021).

**Comment 3**

Line 69: I was wondering how the authors treated missing data due to the presence of sea ice at higher latitudes? This might also affect some of the budget misclosures/uncertainties mentioned farther below in the manuscript.

**Response:** Good point. Although we had not explicitly mentioned this in the manuscript,

our analysis is constrained between 66°S to 66°N of latitude, as can be seen in the global maps. Therefore, the regions where the presence of sea ice might be an issue for satellite altimetry are not included. We now mention the latitudinal limits of the data:

> All data is regridded to 1°x1° map, selected within 66°S to 66°N of latitude, and combined into an ensemble mean, to avoid systematic errors.

**Comment 4**

Line 83: How does that compare the deep ocean contribution from Zanna et al. (2019)?

**Response:** Thank you for your question. The deep ocean contribution based on repeat hydrography estimates are comparable with the estimates from Zanna et al. (2019). However, while the trends from Purkey and Johnson (2010) are statistically significant, the reconstructed deep warming from Zanna et al. (2019) since 1992 is not. A comparison between the Purkey and Johnson (2010) estimates and the ones from Zanna et al. (2019) can be seen in Figure 1 of Zanna et al. (2019).

**Comment 5**

Line 85 following: As mentioned as a general comment, the approach seems to be inconsistent with respect to geocentric sea level as measured by satellites.

**Response:** As we clarified above, we used geocentric GRD fingerprints, which is then consistent with the sea level as measured by satellites.

**Comment 6**

Line 94: It might be good to provide a little more information here, given that this other paper is still under review. I was also wondering how the estimates differ from those in Frederikse et al. (2020)?

**Response:** Thank you for your comment. The paper from which we use the GRD estimates has now been published, hence we think that expanding on the method is not necessary. The main differences from the estimates of Frederikse et al. (2020) is the land mass data sets used as input for the sea-level equation, and the uncertainty characterization. For example, Frederikse et al. (2020) uses a GRACE based reconstruction for terrestrial water storage, while Camargo et al. (2022) uses two hydrological models for it. Also, in Frederikse et al. (2020), they use the spatial patterns of the mass loss over the ice sheets from GRACE to estimate the spatial pattern of mass change over the ice sheets prior to 2002. This approach is different than the ones form Camargo et al. (2022), in which the lack of spatial resolution of the data sets prior to GRACE are incorporated in the uncertainties. In practical terms, the main differences are seen in the uncertainty of fingerprints, while the central estimates (i.e., global mean sea-level change) and the fingerprint patterns are comparable. Additionally, the estimates from Frederikse et al. (2020) are of relative sea-level change, while the ones used here of geocentric sea-level change.

**Comment 7**

Line 174 following: I am wondering how sensitive the two approaches are to temporal filtering? Former assessments such as Thompson and Merrifield (2014) have focused on decadal scales (which is likely more relevant for trends). Did the authors test sensitivity to smoothing? Also, have the time series been deseasonalized before applying the clustering technique?

**Response:**

As the machine learning techniques are used to map coherent regions of similar time sea-level variability, they are expected to be sensitive to the time scales present in the input data set. That is why we decided to use the longest time series record we had, until December 2019, for the clustering, to better resolve the temporal variability and capture better the decadal variability relevant for trend analysis. We did test the clustering using time series until 2016 only, and saw that it was not strongly affected by it, as mentioned in Lines 133-135. We did not, however, perform a direct sensitivity test to temporal smoothing. Note that the inferred SOM temporal patterns of seal level are indeed smoothed during the training process, as neurons are updated according to the characteristic temporal scales of the sea level time series. As expected, if an specific time scale is removed from the input data before the training process (smoothing the time serie), the resulting pattern will not capture this scale. Note also that smoothing/smearing algorithms are needed to create altimetry gridded products from satellite tracks. As a result, sea level data used are already smoothed.

Yes, the time series have been deseasonalized before applying the clustering techniques, as stated in L135.

> **Comment 8**
>
> Line 207: Or atmospheric teleconnections. Not all of them are connected by coasts

**Response:** Thank you for your suggestion. Indeed, they can also represent atmospheric teleconnections. We added this to the sentence:

> areas adjacent to the 'ENSO-tongue' domain, both north and south are clustered together in domain 18 (light blue) or in domain 15 (moss green), indicating how the ENSO signal is propagated through the Pacific, possibly through coastally

trapped waves (Hughes et al., 2019) in the coastal domains (15), or via atmospheric teleconnections.

**Comment 9**

Line 231: does this mean a positive bias?

**Response:** We are not sure what the reviewer means with 'positive bias'. In the referred line, we mention how the residuals decrease when a coarser spatial scale (i.e., $\delta-$MAPS and SOM) is used, comparing to 1 degree resolution. If with the positive bias refers to the fact that in general the altimetry trends were larger than the sum of the budget components (i.e., more positive residuals), this is mentioned in L250. In fact this "bias" is reduced using the machine learning approaches.

**Comment 10**

Line 344: The authors might consider Calafat et al. (2013) and Dangendorf et al. (2014), who initially established that link

**Response:** We have added the references as suggested by the reviewer.

**Comment 11**

Line 349: Southern Hemisphere

**Response:** Corrected.

**References**

Calafat, F. M., Chambers, D. P., & Tsimplis, M. N. (2013). Inter-annual to decadal sea-level variability in the coastal zones of the Norwegian and Siberian Seas: The role of atmospheric forcing. *Journal of Geophysical Research: Oceans*, *118*(3), 1287–1301. https://doi.org/10.1002/jgrc.20106

Camargo, C. M. L., Riva, R. E. M., Hermans, T. H. J., & Slangen, A. B. A. (2022). Trends and uncertainties of mass-driven sea-level change in the satellite altimetry era. *Earth Syst. Dynam. Discuss.*, *13*, 1351–1375. https://esd.copernicus.org/preprints/esd-2021-80/

Dangendorf, S., Calafat, F. M., Arns, A., Wahl, T., Haigh, I. D., & Jensen, J. (2014). Mean sea level variability in the North Sea: processes and implications. *Journal of Geophysical Research: Oceans*, (119), 3868–3882. https://doi.org/10.1002/2014JC009901.Received

Dangendorf, S., Frederikse, T., Chafik, L., Klinck, J. M., Ezer, T., & Hamlington, B. D. (2021). Data-driven reconstruction reveals large-scale ocean circulation control on coastal sea level. *Nature Climate Change*, *11*(6), 514–520. https://doi.org/10.1038/s41558-021-01046-1

Frederikse, T., Landerer, F., Caron, L., Adhikari, S., Parkes, D., Humphrey, V. W., Dangendorf, S., Hogarth, P., Zanna, L., & Cheng, L. (2020). The causes of sea-level rise since 1900. *Nature*, *584*(August). https://doi.org/10.1038/s41586-020-2591-3

Frederikse, T., Riva, R., Kleinherenbrink, M., Wada, Y., van den Broeke, M., & Marzeion, B. (2016). Closing the sea level budget on a regional scale: Trends and variability on the Northwestern European continental shelf. *Geophysical Research Letters*, *43*(20), 10, 810–864, 872. https://doi.org/10.1002/2016GL070750

Frederikse, T., Riva, R. E., & King, M. A. (2017). Ocean Bottom Deformation Due To Present-Day Mass Redistribution and Its Impact on Sea Level Observations. *Geophysical Research Letters*, *44*(24), 12, 306–12, 314. https://doi.org/10.1002/2017GL075419

Frederikse, T., Simon, K., Katsman, C. A., & Riva, R. (2017). The sea-level budget along the Northwest Atlantic coast: GIA, mass changes, and large-scale ocean dynamics. *Journal of Geophysical Research : Oceans*, (122), 5486–5501. https://doi.org/10.1002/2016JC012335.Received

Hughes, C. W., Fukumori, I., Griffies, S. M., Huthnance, J. M., Minobe, S., Spence, P., Thompson, K. R., & Wise, A. (2019). Sea Level and the Role of Coastal Trapped Waves in Mediating the Influence of the Open Ocean on the Coast. *Surveys in Geophysics*, *40*(6), 1467–1492. https://doi.org/10.1007/s10712-019-09535-x

Purkey, S. G., & Johnson, G. C. (2010). Warming of Global Abyssal and Deep Southern Ocean Waters between the 1990s and 2000s: Contributions to Global Heat and Sea Level Rise Budgets. *Journal of Climate*, *23*, 6336–6351. https://doi.org/10.1175/2010JCLI3682.1
deep steric

Rietbroek, R., Brunnabend, S.-E., Kusche, J., Schröter, J., & Dahle, C. (2016). Revisiting the contemporary sea-level budget on global and regional scales. *Proceedings of the National Academy of Sciences*, *113*(6), 1504–1509. https://doi.org/10.1073/pnas.1519132113

Slangen, A. B. A., Adloff, F., Jevrejeva, S., Leclercq, P. W., Marzeion, B., Wada, Y., & Winkelmann, R. (2017). A Review of Recent Updates of Sea-Level Projections at Global and Regional Scales. *Surveys in Geophysics*, *38*(1), 385–406. https://doi.org/10.1007/s10712-016-9374-2

Stammer, D., Cazenave, A., Ponte, R. M., & Tamisiea, M. E. (2013). Causes for Contemporary Regional Sea Level Changes. *Annu Rev Marine Sci*, *5*. https://doi.org/10.1146/annurev-marine-121211-172406

Thompson, P. R., & Merrifield, M. A. (2014). A unique asymmetry in the pattern of recent sea level change. *Geophysical Research Letters*, *41*(21), 7675–7683. https://doi.org/10.1002/2014GL061263

Wang, J., Church, J. A., Zhang, X., Gregory, J. M., Zanna, L., & Chen, X. (2021). Evaluation of the Local Sea-Level Budget at Tide Gauges Since 1958. *Geophysical Research Letters, 48*(20), 1–12. https://doi.org/10.1029/2021GL094502

Zanna, L., Khatiwala, S., Gregory, J. M., Ison, J., & Heimbach, P. (2019). Global reconstruction of historical ocean heat storage and transport. *Proceedings of the National Academy of Sciences of the United States of America, 116*(4), 1126–1131. https://doi.org/10.1073/pnas.1808838115

---

## Author Comment (AC4)

**Authors' Response to Reviewer 3**

> **General Comment.**
>
> Global sea level budgets are examined using two machine learning techniques. Through identifying regions of similar sea level variability, the authors examined sea level budget in different basins of the world oceans. It is found that for most of the ocean regions, sea level variation can be explained using steric height changes and mass transport between ocean and land. But for some highly dynamic regions, the sea level budget closure may be affected by the mass redistribution associated with strong western boundary currents. All these make sense to this Reviewer. This is an excellent example of SOM application in oceanography and climate research community. I would like to recommend the manuscript be accepted after some minor revision. Specific comments are listed as follows.

**Response:**

Dear Reviewer,

Thank you for your feedback and positive review. We have addressed all the issues item by item as follows.

Kind regards,

Carolina Camargo, on behalf of the authors

**Comment 1**

Pioneer work on SOM analysis of sea level variability should be properly mentioned. These include the first time SOM analysis of the satellite altimetry data (Liu et al., 2008), and the dual-SOM applications including the regionalizing of sea level variability in the Gulf of Mexico (Liu et al., 2016). It would be good to add the following information to the paragraph explaining the SOM (L138 - L156) or the Introduction part (L44-45):

"SOM has been used to extract patterns of sea level variability from satellite altimetry data (Liu et al., 2008; Nickerson et al., 2022; Weisberg & Liu, 2017). Dual-SOM application has been proposed to analyse sea level data, one focused on the characteristic spatial patterns, and the other focused on the characteristic time series, using sea level in the Gulf of Mexico as an example (Liu et al., 2016). The latter resulted in regionalizing the sea level variability, and is pursued here in this study to analyse global sea level data."

**Response:** We have added the information about pioneer work on SOM on the paragraph explaining the SOM

> The ability of SOM to extract patterns of sea level variability from satellite altimetry data has been shown in previous works (e.g., Hardman-Mountford et al., 2003; Iskandar, 2009; Liu et al., 2016; Liu et al., 2008; Nickerson et al., 2022; Weisberg & Liu, 2017). To analyse sea level data, SOM can be applied either on the spatial domain, focusing on the characteristic spatial patterns, or on the time domain, focusing on the characteristic time series (Liu et al., 2016). The latter results in regionalizing the sea-level variability, and is pursued here in this study to analyse global sea level data.

> **Comment 2**
>
> L361-L362 indicate the challenges of sea level budget in coastal regions. This is true, as coastal ocean dynamics of sea level (e.g., Liu & Weisberg, 2007) are more complicated than that of deep ocean, and key dynamics may not be properly represented in the global data. It would be good to add a sentence to L364 about the sea level budget issues for coastal regions: "Note that sea level budget for coastal regions is more challenging (Liu & Weisberg, 2007) with some of the dominant coastal ocean dynamics are not properly represented in the global data sets."

**Response:** Thank you for the comment. Indeed, the sea-level budget in coastal regions is more challenging. We added a sentence about this, as suggested by the reviewer:

> Note that the sea-level budget in coastal regions is more challenging (Dangendorf et al., 2021), since some of the dominant coastal ocean dynamics are not properly represented in the global data sets (Liu & Weisberg, 2007).

> **Comment 3**
>
> Throughout the manuscript, "sea-level" should be changed to "sea level" —— no hyphen.

**Response:** We appreciate the suggestion. There is no clear consensus if sea level should be hyphened or not. We hyphenate sea level when it is used as a compound modifier and not when used a *noun* phrase. For example: "Global sea level rose over the past century."; "What causes sea-level rise?"

**Comment 4**

The abbreviations of "sea level change" and "sea level budget" are not necessary at all. They do not save much space in text, rather they may cause inconveniences to readers, as readers may need to go back to search what they stand for, particularly for the case of many other acronyms are used later.

**Response:** We understand that acronyms can cause confusion. We have removed the abbreviations of SLC and SLB for "sea-level change" and "sea-level budget", respectively.

**Comment 5**

L23, it would be good to provide an example, Chambers et al. (2017), for this sentence.

**Response:** We added the reference example to the sentence:

> The attribution of sea-level change to its different drivers is typically done using a sea-level budget approach (Cazenave et al., 2018; Chambers et al., 2017).

**Comment 6**

L90, GRD is not defined.

**Response:** Thank you for calling our attention. It should have been defined in the previous paragraph. We added GRD definition:

> $\eta_{GRD}$, the Gravitational, Rotational and viscoelastic Deformation (GRD) response of the Earth to ...

**Comment 7**

Line 139, it would be good to insert a sentence to mention the powerfulness of the SOM technique: "It was demonstrated to be more powerful than conventional feature extraction methods (e.g., Liu & Weisberg, 2005).

**Response:** Thank you for the suggestion. We added the mention of the advantage of SOM over the other techniques:

SOM (Kohonen, 1982) is a feature extraction and classification method based on an unsupervised neural network (Liu et al., 2006), which was demonstrated to be more powerful than conventional feature extraction methods (e.g., Liu & Weisberg, 2005).

**References**

Cazenave, A., Meyssignac, B., Ablain, M., Balmaseda, M., Bamber, J., Barletta, V., Beckley, B., Benveniste, J., Berthier, E., Blazquez, A., Boyer, T., Caceres, D., Chambers, D., Champollion, N., Chao, B., Chen, J., Cheng, L., Church, J. A., Chuter, S., . . . Wouters, B. (2018). Global sea-level budget 1993-present. *Earth System Science Data*, *10*(3), 1551–1590. https://doi.org/10.5194/essd-10-1551-2018

Chambers, D. P., Cazenave, A., Champollion, N., Dieng, H., Llovel, W., Forsberg, R., von Schuckmann, K., & Wada, Y. (2017). Evaluation of the Global Mean Sea Level Budget between 1993 and 2014. *Surveys in Geophysics*, *38*(1), 309–327. https://doi.org/10.1007/s10712-016-9381-3

Dangendorf, S., Frederikse, T., Chafik, L., Klinck, J. M., Ezer, T., & Hamlington, B. D. (2021). Data-driven reconstruction reveals large-scale ocean circulation

control on coastal sea level. *Nature Climate Change*, *11*(6), 514–520. https://doi.org/10.1038/s41558-021-01046-1

Hardman-Mountford, N. J., Richardson, A. J., Boyer, D. C., Kreiner, A., & Boyer, H. J. (2003). Relating sardine recruitment in the Northern Benguela to satellite-derived sea surface height using a neural network pattern recognition approach. *Progress in Oceanography*, *59*(2-3), 241–255. https://doi.org/10.1016/j.pocean.2003.07.005

Iskandar, I. (2009). Variability of Satellite-Observed Sea Surface Height in the Tropical Indian Ocean: Comparison of Eof and Som Analysis. *MAKARA of Science Series*, *13*(2), 173–179. https://doi.org/10.7454/mss.v13i2.421

Kohonen, T. (1982). Self-organized formation of topologically correct feature maps. *Biological Cybernetics*, *43*(1), 59–69. https://doi.org/10.1007/BF00337288

Liu, Y., & Weisberg, R. H. (2005). Patterns of ocean current variability on the West Florida Shelf using the self-organizing map. *Journal of Geophysical Research: Oceans*, *110*(6), 1–12. https://doi.org/10.1029/2004JC002786

Liu, Y., & Weisberg, R. H. (2007). Ocean currents and sea surface heights estimated across the west Florida shelf. *Journal of Physical Oceanography*, *37*(6), 1697–1713. https://doi.org/10.1175/JPO3083.1

Liu, Y., Weisberg, R. H., & Mooers, C. N. (2006). Performance evaluation of the self-organizing map for feature extraction. *Journal of Geophysical Research: Oceans*, *111*(5), 1–14. https://doi.org/10.1029/2005JC003117

Liu, Y., Weisberg, R. H., Vignudelli, S., & Mitchum, G. T. (2016). Patterns of the loop current system and regions of sea surface height variability in the eastern Gulf of Mexico revealed by the self-organizing maps. *Journal of Geophysical Research: Oceans*, *121*(5), 2347–2366. https://doi.org/10.1002/2015JC011493.Received

Liu, Y., Weisberg, R. H., & Yuan, Y. (2008). Patterns of upper layer circulation variability in the South China Sea from satellite altimetry using the self-organizing map. *Acta Oceanologica Sinica*, *27*(SUPPL.), 129–144.

Nickerson, A. K., Weisberg, R. H., & Liu, Y. (2022). On the Evolution of the Gulf of Mexico Loop Current Through Its Penetrative, Ring Shedding and Retracted

States. *Advances in Space Research, 69*(11), 4058–4077. https://doi.org/10.1016/ j.asr.2022.03.039

Weisberg, R. H., & Liu, Y. (2017). On the Loop Current Penetration into the Gulf of Mexico. *Journal of Geophysical Research: Oceans, 122*(12), 9679–9694. https: //doi.org/10.1002/2017JC013330

---

## Author Comment (AC5)

**Authors' Response to Reviewer 4**

> **General Comment.**
>
> The authors offer a new perspective on regional sea level budget (SLB) closure by focusing on two machine learning (ML) algorithms. They show how self organizing maps (SOM) and d-MAPS allow to identify spatially coherent regions and (i) close the SLB in most of those regions and (ii) further reduce the uncertainties otherwise present when using the gridded data alone. Importantly, by focusing on two ML tools they are able to demonstrate the robustness of their conclusions under the models architecture considered. This is a good, well written paper and it further demonstrate the benefit of focusing on coherent patterns rather than gridded data. I recommend publication after some minor revisions.

**Response:**

Dear Reviewer,

Thank you for your feedback and positive review. We have addressed all the issues item by item as follows.

Kind regards,

Carolina Camargo, on behalf of the authors

> **Comment 1**
>
> Section 2.1. Is terrestrial water storage included in the sea level budget? If not, can the authors give an explanation on why it was not considered?

**Response:** Yes, the terrestrial water storage is included in the sea-level budget, in the GRD component. We added a sentence to Section 2.1 to clarify which terms are included in the GRD component:

For the GRD component, we use the estimates from Camargo et al. (2022), which includes the geocentric sea level response to changes on the Antarctic and Greenland ice sheets, glaciers and terrestrial water storage.

**Comment 2**

Figure 1. In a recent paper, Wang et al. (2021) closed the sea level budget at tide gauges locations. In Figure 1 of Wang et al. (2021), the authors show different components of sea level (SL) ranging from stereodynamic SL to GIA, Glaciers etc. Can the authors explain possible differences between their Figure 1 and the one in Wang et al. (2021). In any case the authors should at least cite that work.

**Response:** Thank you for your comment. The first and main difference between our estimates and the ones from Wang et al. (2021) is the time period: Figure 1 of Wang et al. (2021) shows trends from 1958-2015, and our Figure 1 from 1993-2016. This will result in different global mean trends and patterns shown in the figures. For example, their sterodynamic trend has a global mean trend of 0.7 mm/yr, while we have 1.55 mm/yr for the steric component. Another difference is in the data sets they use for their estimates, for example they use the average of three global mean thermosteric data sets, while we use the ensemble mean of fifteen regional steric data sets (including both the salinity and temperature effects). Finally, their barystatic-GRD fingerprints show relative sea-level change, since they are comparing it with tide gauges, while we use absolute sea-level change fingerprints. Nevertheless, we agree that the work of Wang et al. (2021) should be acknowledged in our manuscript. We have added a reference to their work in the introduction:

The sea-level budget has also been analysed for individual coastline stretches characterized by coherent variability (Dangendorf et al., 2021; Frederikse et al., 2016; Frederikse et al., 2017; Rietbroek et al., 2016), and at individual tide gauges (Wang et al., 2021).

**Comment 3**

It would be clearer if SOM and d-Maps are introduced in two different subsections or paragraphs (Section 2.3.1 and 2.3.2)

**Response:** Thank you for your suggestion. We added subsections to the paragraphs introducing SOM and d-Maps.

**Comment 4**

I understand that in both SOM and d-Maps, domains are identified after removing the seasonal cycle and trends. This is reasonable. It is my understanding though that after the domain identification step, the time series considered in each domain are averaged time series with seasonality and trends included. Is this correct? This should be clearly stated in the manuscript and it is missing at the moment (my bad in case I missed it).

**Response:** Indeed, after the domain identification, for the budget analysis the time series include seasonality and trends. To clarify it, we added this information in Section 2.3:

> We pre-process the input data by removing the global mean trend, seasonality and by applying a spatial Gaussian filter of 300km half-width to remove small scale variability. Note that, after the domains identification, for the budget analysis, global mean trend, seasonality and small scale variability are included in the time series.

and also in Section 2.2:

> Note that, unlike for the identification of the domains (Section 2.3), the time series used to estimate trends and uncertainties include seasonality and global mean trends.

**Comment 5**

Section 4.1. The budget is closed in 77/92 d-Maps domains. I wonder if the remaining 15 domains where the budget is not closed are mainly found in the Southern Ocean. In that region I see lots of very small regions which could be just noise. If that is the case I suggest the author to add this in the paper.

**Response:** Not all of those regions are in the Southern Ocean, as shown in the Figure below. Most of them are small regions, in which probably the non-closure of the budget is related to noise. Other regions might be due to the dynamic influence (e.g., in the Malvinas Confluence zone), or because they are regions covering continental shelves and close to the coast. We have added this information to Section 5.

Figure 1: δ-MAPS regions in which the sea-level budget is not closed.

**Comment 6**

Figure 4. "Sea level budget trends (mm/yr) for (a) d-MAPS ad (b) SOM" I think it should be the opposite: Sea level budget trends (mm/yr) for (a) SOM and (b) d-MAPS.

**Response:** Thank you for pointing this out, indeed it should be (a) for SOM and (b) for d-MAPS. We have corrected it accordingly.

**References**

Camargo, C. M. L., Riva, R. E. M., Hermans, T. H. J., & Slangen, A. B. A. (2022). Trends and uncertainties of mass-driven sea-level change in the satellite altimetry era. *Earth Syst. Dynam. Discuss.*, *13*, 1351–1375. https://esd.copernicus.org/preprints/esd-2021-80/

Dangendorf, S., Frederikse, T., Chafik, L., Klinck, J. M., Ezer, T., & Hamlington, B. D. (2021). Data-driven reconstruction reveals large-scale ocean circulation control on coastal sea level. *Nature Climate Change*, *11*(6), 514–520. https://doi.org/10.1038/s41558-021-01046-1

Frederikse, T., Riva, R., Kleinherenbrink, M., Wada, Y., van den Broeke, M., & Marzeion, B. (2016). Closing the sea level budget on a regional scale: Trends and variability on the Northwestern European continental shelf. *Geophysical Research Letters*, *43*(20), 10, 810–864, 872. https://doi.org/10.1002/2016GL070750

Frederikse, T., Simon, K., Katsman, C. A., & Riva, R. (2017). The sea-level budget along the Northwest Atlantic coast: GIA, mass changes, and large-scale ocean dynamics. *Journal of Geophysical Research : Oceans*, (122), 5486–5501. https://doi.org/10.1002/2016JC012335.Received

Rietbroek, R., Brunnabend, S.-E., Kusche, J., Schröter, J., & Dahle, C. (2016). Revisiting the contemporary sea-level budget on global and regional scales. *Proceedings of the National Academy of Sciences*, *113*(6), 1504–1509. https://doi.org/10.1073/pnas.1519132113

Wang, J., Church, J. A., Zhang, X., Gregory, J. M., Zanna, L., & Chen, X. (2021). Evaluation of the Local Sea-Level Budget at Tide Gauges Since 1958. *Geophysical Research Letters*, *48*(20), 1–12. https://doi.org/10.1029/2021GL094502